# MorphoNet 2.0: An innovative approach for qualitative assessment and segmentation curation of large-scale 3D time-lapse imaging datasets

**Benjamin Gallean[1,2,3†], Tao Laurent[1†], Kilian Biasuz[2], Ange Clement[1], Noura Faraj[1], Patrick Lemaire[2], Emmanuel Faure[1]\***

[1]Laboratoire d'informatique, de robotique et de microélectronique de Montpellier, LIRMM, Université de Montpellier, CNRS, Montpellier, France; [2]Centre de Recherche de Biologie cellulaire de Montpellier, CRBM, Université de Montpellier, CNRS, Montpellier, France; [3]Montpellier Ressources Imagerie, Biocampus, Université de Montpellier, CNRS, INSERM, Montpellier, France

**\*For correspondence:**
emmanuel.faure@lirmm.fr

[†]These authors contributed equally to this work

**Competing interest:** The authors declare that no competing interests exist.

## eLife Assessment

This **important** work presents technical and conceptual advances with the release of MorphoNet 2.0, a versatile and accessible platform for 3D+T segmentation and analysis. The authors provide **compelling** evidence across diverse datasets, and the clarity of the manuscript together with the software's usability broadens its impact. Although the strength of some improvements is hard to fully gauge given sample complexity, the tool is a significant step forward that will likely impact many biological imaging fields.

**Abstract** Thanks to recent promising advances in AI, automated segmentation of imaging datasets has made significant strides. However, the evaluation and curation of 3D and 3D+t datasets remain extremely challenging and highly resource-intensive. We present MorphoNet 2.0, a major conceptual and technical evolution designed to facilitate the segmentation, self-evaluation, and correction of 3D images. The application is accessible to non-programming biologists through user-friendly graphical interfaces and works on all major operating systems. We showcase its power in enhancing segmentation accuracy and boosting interpretability across five previously published segmented datasets. This new approach is crucial for producing ground-truth datasets of discovery-level scientific quality, critical for training and benchmarking advanced AI-driven segmentation tools, as well as for competitive challenges.

## Introduction

Recent advances in optical time-lapse microscopy now enable the 3D capture of life's temporal dynamics, from molecular assemblies to entire organisms (*McDole et al., 2018*), in cells, embryos, and aggregates. These technologies have thus become crucial for cell and developmental biology. The automated segmentation and tracking of these complex datasets is, however, necessary for their interpretation and quantification (*McDole et al., 2018*).

Recent progress in the 3D segmentation of animal and plant cells (*Pachitariu and Stringer, 2022*), or organelles (*Cutler et al., 2022*) has made this process highly efficient, despite the persistence

of algorithm-dependent systematic errors (*Liu et al., 2023*). The processing of time-lapse datasets, however, remains challenging. The accumulation over hundreds of time points of even a few residual segmentation errors can strongly affect data interpretation and notably disrupt long-term cell tracking. Following the excitement generated by this new generation of imaging systems, which offers the ability to image living systems at unprecedented spatial and temporal scales, a glass ceiling has now been reached due to the quality of these reconstructed data, which fails to scale to real research exploitations.

Beyond the reconstruction requirements of individual research projects, proper metrics are essential for evaluating the performance of bioimage analysis algorithms (*Nature Methods, 2024*). However, the training and benchmarking of next-generation AI-based segmentation tools fundamentally depend on the availability of accurate ground truth data (*Maška et al., 2023*). The critical bottleneck is the lack of a massive amount of 3D high-quality reconstructed data to train these systems.

We have decided to take on this challenge and develop a tool to produce high-precision and high-quality 3D datasets for AI. Two major barriers must be overcome to reach this milestone. The first concerns the **evaluation** of dataset reconstructions: how can we measure the accuracy of segmentations obtained by automated algorithms for these complex 3D and 3D+t datasets? The second focuses on the **curation** of reconstructed data: how can we correct and significantly enhance segmentations to achieve reliable reconstructions that are essential for major scientific breakthroughs?

2D imaging has long been a cornerstone of biological research, driving significant scientific discoveries with visual validation being largely adequate, as it could be performed directly on a 2D screen. The traditional method involves visually comparing the reconstruction by overlaying it with the acquired data, as seen in classic software like ImageJ (*Schneider et al., 2012*), Napari (*Sofroniew, 2022*), or Ilastik (*Berg et al., 2019*). When discrepancies arise, experts can perform curation and manually adjust the pixel values to align with the desired class (e.g. a cell).

In the context of 3D and 3D+t data, the technique encounters significant obstacles that hinder reliable scientific analysis. The 3D structure of images, inherently unsuited to the 2D digital environment, demands numerous compromises that restrict their interpretability. Projecting 3D objects onto a 2D screen causes data masking, requiring experts to perform extensive manipulations to verify accuracy. Additionally, the high number and density of 3D objects further complicate the process, as visual interference from overlapping objects increases during these adjustments. As a result, voxel-level (3D pixel) curation becomes almost unfeasible using conventional methods, compelling experts to revert to working in 2D. This approach becomes very time-consuming and, therefore, does not allow to obtain a satisfactory reconstruction quality for scientific use. A final obstacle is caused by the much larger size of 3D datasets, which significantly lengthens processing time for each curation action. This computational burden becomes daunting for 3D+t time-lapse datasets, as error propagation between consecutive time points rapidly increases the number of segmentation and object tracking errors needing correction. Restricting the application of advanced image processing tasks to subsets of the image may alleviate this issue.

An unsupervised objective assessment can be derived from a priori knowledge of expected spatial (e.g. smoothness of contours, shape regularity, volume…) or temporal (e.g. stability and smooth evolution, lifetime of objects…) features of the objects (*Correia and Pereira, 2002*). Features of the object boundaries (e.g. contrast between inside and outside of an object) can also be computed (*Correia and Pereira, 2002*). More sophisticated strategies have been proposed when no statistically relevant a priori knowledge can be drawn (*Valindria et al., 2017*). Since curation is typically carried out by expert biologists with limited programming skills, the computation of image features and the projection of relevant metrics onto individual segmented objects should be accessible through user-friendly interfaces. An alternative approach for interacting with 3D segmented images is, therefore, necessary.

## Results

We previously developed a novel concept of morphogenetic web browser, MorphoNet (*Leggio et al., 2019*), to **visualize** and **interact** with 3D+t segmented datasets through their meshed representations. This user-friendly web solution required no installation and was suited for datasets of moderate size (up to a thousand cells over a few hundred time points). Since its release, this platform has been used in a variety of morphological studies in plant and animal systems (*Manni et al., 2022*; *Chung et al., 2023*) and has been instrumental to interpret the relationship between gene expression

and complex evolving cell shapes (*Refahi et al., 2021*; *Guignard et al., 2020*; *Dardaillon et al., 2020*). This web version was restricted to visualizing and interacting with segmented datasets through their precomputed 3D dual meshes. This approach offered significant benefits in terms of online 3D rendering efficiency and data management but it also came with limitations. For example, meshes are simplified representations of the segmented object that may not capture fine details of the original volumetric data and that cannot be validated without comparison to the original raw intensity images. Meshed representations can also make it challenging to perform the precise computations required for detailed geometric and topological analyses (*Sophie et al., 2024*). Exploration of larger datasets was nonetheless constrained by the computational resource limitations of internet browsers.

We present here MorphoNet 2.0, a major conceptual evolution of the platform, designed to offer powerful tools for the **reconstruction**, **evaluation**, and **curation** of 3D and 3D+t datasets. This innovative approach leverages the richness and redundancy of information embedded within these complex datasets. It addresses the two previously identified challenges: **evaluating** 3D segmented data without relying on manual ground truth by fully harnessing the data's richness, and enabling semi-automated **curation** of 3D data, now achievable with just a few clicks. This enables us to achieve data quality at a level suitable for scientific discovery. We showcase the impact of these advancements by revisiting and enhancing five published animal and plant datasets previously regarded as ground truth (*Maška et al., 2023*).

We feature a new standalone application running on all major operating systems that exploits the resources of the user's local machine to explore, segment, assess, and manually curate very large 3D and 3D+t datasets. Like the web version, the MorphoNet application uses the power and versatility of

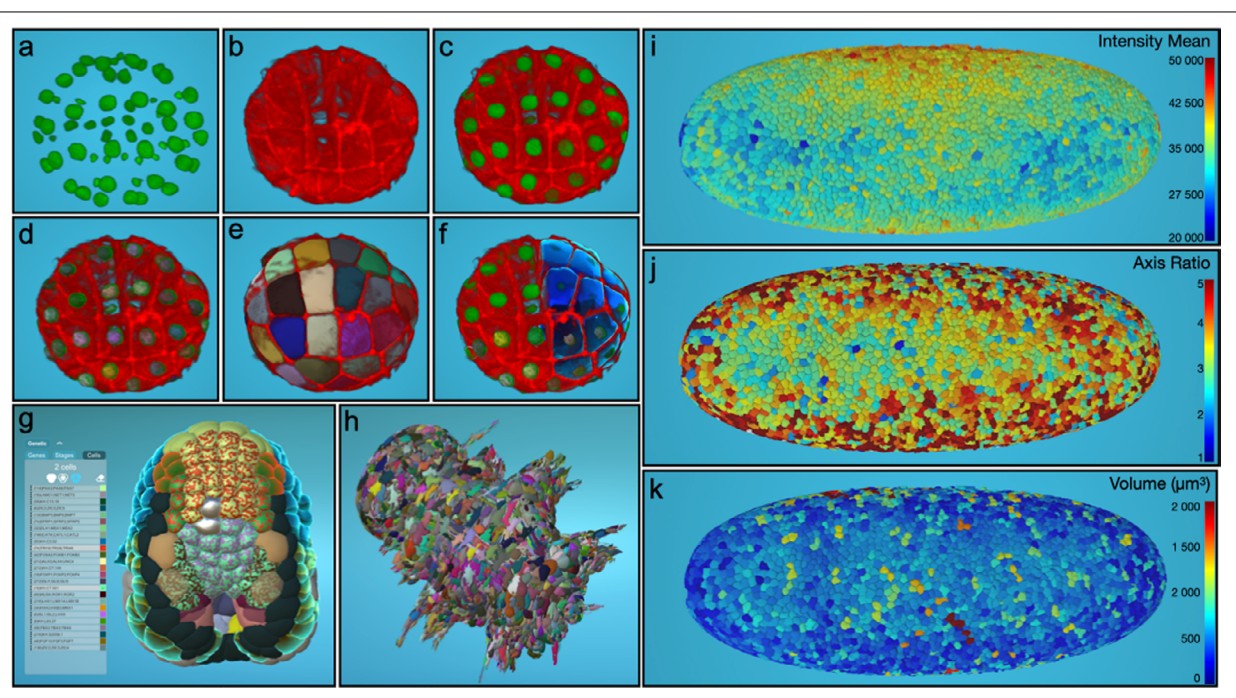

**Figure 1.** Illustration of the visualization of datasets of various complexity and nature in the MorphoNet standalone application. (**a–f**) Visualization of a 64 cell stage Phallusia mammillata embryo with labeled cell nuclei and cell membranes. (**a-c**) Intensity images showing the nuclei (**a**), the membranes (**b**) or both (**c**). (**d**) Same as **c** with additional nuclei segmentation obtained with the Binarize plugin (see Materials and methods for the full description of curation plugins). (**e**) Same as **b** with additional membrane segmentation obtained with the Cellpose 3D plugin. (**f**) Same as **c** with a combination of several rendering possibilities of cell and nuclei segmentations. (**g**) Multi-colored shaders allow the simultaneous visualization of the expression patterns of multiple genes extracted from the ANISEED (*Dardaillon et al., 2020*) database and of tissue fate information. Ascidian embryo (*Guignard et al., 2020*) at stage 15 (mid neurula); cells with a single color are colored with larval tissue fate; multi-colored cells are colored with both larval tissue fate and the expression of selected genes. (**h**) Visualization of a 6 days post-fertilization *Platynereis dumerilii* embryo (*Vergara et al., 2021*) imaged by whole-body serial block face scanning electron microscopy followed by the automated whole-cell segmentation of 16,000 cells. (**i-k**) Visualization of a cell cycle 14 *Drosophila melanogaster* embryo imaged with SiMView microscopy and segmented with RACE (*Stegmaier et al., 2016*). (**i**) Projection on each segmented cell of the mean image intensity. (**j**) Projection on each segmented cell of the ratio between the length of the major and the minor axes of the ellipse that has the same normalized second central moments as the segmented cell. (**k**) Projection of the cell volume.

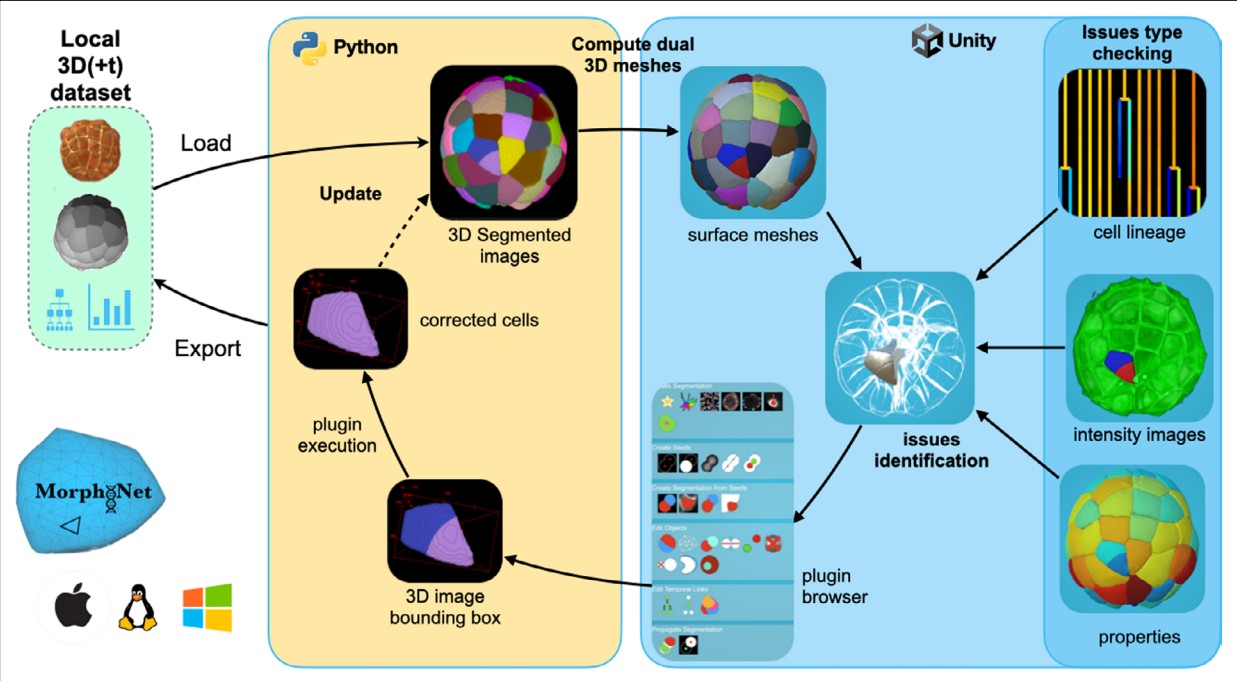

**Figure 2.** MorphoNet Standalone Schema. From the local data (loaded from the green box), the MorphoNet Standalone application first computes the dual meshes for each of the segmented objects (in the module python in the yellow box). Then, using the 3D viewer (in the blue box), users identify detection, segmentation, or tracking issues using, if necessary, the cell lineage information, the raw images, and/or properties computed on the segmented objects. Errors are then corrected by choosing and executing the appropriate image processing plugin from the curation menu. Finally, new meshes are computed from the result of the plugin execution to update the visualization.

the Unity game engine and includes its main features (*Leggio et al., 2019*). By overcoming web-based limitations, the application can handle more complex datasets, including heavier segmented voxel images and up to several tens of thousands of objects on a research laptop (*Figure 1h*, and *Supplementary file 1*). The standalone application allows users to directly explore private data stored on their own computer, without need for an upload to the MorphoNet server. Subsequent data sharing with other researchers or a wider public in an open science process is facilitated by the MorphoNet server upload functions of the standalone.

MorphoNet is designed with a strong emphasis on user-friendliness, making its use highly intuitive for experimental biologists with no coding experience (See Videos in Materials and Methods). Every feature and interface element has been crafted to ensure a smooth, straightforward user experience, allowing both beginners and experts to utilize its capabilities efficiently. Additionally, the solution is fully open-source allowing bio-image analysis to develop their own features (See *Supplementary file 2*).

One of the key features of this new standalone application is the automatic integration of a duality between the 3D segmented images and their corresponding meshes (*Figure 2*). This feature is achieved by linking Python, dedicated to high-performance image processing, with the Unity Game Engine, fully optimized for seamless interaction with the dual meshes. This integration enables access to state-of-the-art reconstruction (*Pachitariu and Stringer, 2022*; *Weigert et al., 2020*) tools, including those leveraging AI libraries, while also addressing the challenges of interacting with 3D images by harnessing the powerful interactive capabilities of game engines.

To ensure users can consistently assess segmentation quality, raw intensity images are also available by superimposition. This provides access to the entire workflow, from raw image acquisition to meshed segmentation, allowing simultaneous exploration and visualization of both meshed and voxel images (*Figure 1a–f*).

We also leveraged Unity's scene management tools to implement simultaneous visualization of multiple scenes. This feature enables the display of interactive cell lineages in a dedicated, fully connected window, offering essential insights into cellular trajectories within 3D+t datasets.

## Automatic error detection

Efficient dataset curation requires the rapid identification of segmentation errors in large 3D or time-lapse datasets containing thousands of segmented objects. MorphoNet 2.0 tackles this critical challenge by introducing unsupervised metrics for objective, automated assessment, eliminating the subjectivity, inefficiency, and inconsistency inherent in manual visual inspections. These metrics leverage prior knowledge of generic data properties, such as shape regularity, contour smoothness, and temporal stability of shapes and volumes - to deliver scalable and consistent analysis. Beyond improving curation efficiency, these metrics offer quantitative scores that enable systematic comparisons across datasets, experiments, and tools. Integrated into MorphoNet, they support benchmarking, algorithm refinement, and reproducibility. Using the scikit-image library (*van der Walt et al., 2014*), we automatically compute object properties from both segmented and intensity images. These properties quickly help identify outliers, which often correspond to segmentation errors. While the traditional approach uses some metrics to evaluate global distributions, it is crucial for curation purposes to apply these properties at the level of each individual segmented object. Thus, these properties, including shape metrics like volume, convexity, and elongation, and intensity metrics, such as mean voxel intensity within or around segmented objects, can be easily projected onto meshed objects for visualization (*Figure 1i–k*).

Segmentation quality assessment is further enhanced by calculating three categories of properties for each dataset: (1) Morphological features – including volume, convexity, roughness, and elongation, (2) Intensity-based measurements – within and around each object in the original acquisition images, such as mean intensity, homogeneity, or deviation at segmentation borders; (3) Temporal features, such as object lifetime or cell lineage distances (*Guignard et al., 2020*).

By projecting these values onto segmented object meshes, outliers are readily identified as prime candidates for curation. These values can also be exported and plotted to reveal distributions, providing an unsupervised assessment of overall segmentation quality. This process is exemplified in various use cases described below.

To evaluate the relevance of these unsupervised metrics, we compared their distributions in the published and curated datasets against a manually annotated gold standard in Use Case 1. The curated segmentations exhibited higher Intersection over Union (IoU) scores (*Figure 3—figure supplement 2*). Although these unsupervised metrics are not flawless predictors of segmentation accuracy, they show a strong correlation with standard quality indicators, such as IoU. This confirms their practical utility as reliable proxies for segmentation quality and valuable tools to guide efficient manual curation.

A key limitation of relying solely on unsupervised metrics is the risk of circular logic: corrections or model training based on masks deemed 'good' by these metrics may promote homogeneity without improving biological accuracy. This can lead to segmentations optimized for metric conformity rather than real quality. To avoid this, MorphoNet's metrics are intended as guidance tools for flagging candidate errors, which should then be validated through expert curation, ground truth comparisons, or biological plausibility.

## Biocuration

The execution time of algorithms in 3D image processing limits the optimization of their parameters. Additionally, the inherent heterogeneity within 3D images frequently prevents achieving consistently high-quality results across the entire dataset. To tackle these challenges, we defined in Morphonet the notion of multi-type representation layers, able to combine and interoperate 3D image intensity layers with mesh-based layers. This dual representation enables seamless interaction with meshes, facilitating the rapid identification and precise selection of objects requiring curation. By focusing processing efforts on these selected regions rather than the entire image, MorphoNet significantly reduces the time and effort required, streamlining the curation workflow for complex 3D+t datasets.

The meshed versions of the segmentations are used for 3D rendering, for the identification and selection of objects needing curation, and to launch the needed curation algorithms, which are performed on the segmented images (*Figure 2*). A significant acceleration of operations on 3D images is achieved by targeting the processing activities on a subpart of the image, for instance, on the bounding box of an object or of a set of objects. This approach enables results to be generated in just seconds for most tasks, even with highly complex 3D data. Given the heterogeneity within

3D images, it is challenging to find a single set of parameters that works effectively across the entire image. This approach makes it easy to run image processing algorithms with different parameters in different subregions of the image.

Editions of objects necessitating curation can then proceed using dedicated plugins. The curation module of MorphoNet has an expandable open-source Python plugin architecture and provides user-friendly graphical interfaces accessible to experimental biologists with limited programming skills. To support advanced image processing techniques powered by deep learning, which have an increasing impact on image analysis, the default installation package includes training libraries like Scikit-learn, PyTorch, and TensorFlow. Plugins can take the raw intensity image into account to perform 3D image processing tasks, as exemplified by the integration of popular segmentation tools, such as Stardist (*Weigert et al., 2020*) and Cellpose (*Pachitariu and Stringer, 2022*). The full list of plugins can be found in the Materials and Methods. Users and developers can expand the current plugin list by creating their own through an easy-to-use Python template (see *Supplementary file 2*).

To help the user identify suitable plug-ins, we grouped them into seven categories. The first five are used to create or edit segmentations of 3D datasets, the last two are dedicated to the temporal propagation of corrections in 3D+t datasets. All categories will be exemplified in the five use cases below. Figure 8 in Materials and methods illustrates how the majority of segmentation and tracking errors can be corrected using plugins, single-handedly or in combination. For example, an under-segmentation can be corrected by running locally the *Cellpose* plugin as in use case 2 or the Temporal Propagation plugin (*Prope*), as in use case 5.

1. *De novo Segmentation* plugins: these plugins create a de novo segmentation from the original intensity images (or from a specified region of interest of the intensity images). This category includes segmentation tools, such as Cellpose 3D or Stardist, as well as simpler intensity thresholding. As described below, MorphoNet plugins allow to specifically target the application of these tools to error-rich regions.
2. *De novo Seed* plugins: these plugins automatically create seeds (i.e. 3D points in the 3D images), which can then be used to initiate segmentation tasks. Additionally, seeds can also be added manually using the interface.
3. *Segmentation from Seeds* plugins: these plugins are used to perform local segmentation using predefined seeds, for example, by using a watershed algorithm.
4. *Segmentation correction* plugins: these plugins are used to correct the main classes of segmentation errors on selected segmented objects. This includes the fusion of over-segmented cells, the resolution of small artifactual cells, or the splitting of under-segmented cells.
5. *Shape transform* plugins: these plugins are used to edit the shape of selected segmented objects. This includes classical morphological operators (such as dilate, erode, etc.) or the manual deformation of a cell's shape.
6. *Propagate Segmentation* plugins: these plugins are used to propagate an accurate segmentation at a specific time point to erroneous segmentations at later or earlier time points. Examples: Propagate eroded cell masks to correct under-segmentation errors.
7. *Edit Temporal Links* plugins: these plugins are used to curate cell lineage trees by creating or editing temporal information. Examples: Create temporal links using the spatial overlap between homologous objects at successive time points.

## Use cases

To highlight the efficiency of MorphoNet plugins in detecting and correcting the main types of errors across a broad range of organisms, we present five examples from previously published fluorescent live imaging datasets, representing various levels of quality and complexity. These examples collectively showcase MorphoNet's powerful segmentation quality assessment and error detection capabilities in different contexts, ranging from low-quality nuclear segmentations to high-quality whole-cell segmentations. While these examples frequently involve the use of Cellpose, the current state-of-the-art tool for 3D dataset segmentation, the focus is not on emphasizing the tool's power but on demonstrating how advanced tools can be enhanced through integration into MorphoNet's sophisticated graphical plugin interfaces.

The first use-case introduces two unsupervised metrics for assessing dataset quality in the absence of ground truth on a time-lapse light-sheet microscopy movie of a developing *Tribolium castaneum* embryo. Using these metrics, we demonstrate the power of an iterative segmentation and training

strategy. By applying a Cellpose model, trained on a curated subpopulation, to the entire dataset, we improve segmentation quality.

The second use-case presents unsupervised whole-cell morphological metrics to assess the quality of several independent segmentations of a time-lapse confocal movie of a developing starfish embryo, up to the 512 cell stage, with fluorescently labeled cell plasma membranes. Originally, a bio-image analyst segmented the movie using a complex workflow. We demonstrate that running an imported, custom-trained Cellpose 3D model through MorphoNet's graphical interfaces greatly increases the number of cells that can be successfully segmented.

The third use-case focuses on detecting and resolving regions of under-segmented cells in a confocal 3D dataset of *Arabidopsis thaliana* plant apical shoot meristem, with fluorescent plasma membrane and whole-cell segmentation of 1800 cells. It demonstrates the identification of a large region with heavily under-segmented cells, which is successfully addressed by targeting Cellpose to the specific region of interest.

The fourth use-case addresses the correction of low-quality automated cell nuclei segmentation in a time-lapse confocal *Caenorhabditis elegans* embryo with poor-quality nuclei labeling up to the 350 cell stage, despite accurate manual tracking. It demonstrates how object edition plugins can be used to correct individual nuclei segmentations and propagate these corrections over time.

The final use-case illustrates the efficient detection and correction in the cell lineage of rare residual errors that escaped previous scrutiny in a published time-lapse multiview light-sheet imaging dataset of a developing *Phallusia mammillata* embryo. The dataset features labeled cell membranes and high-quality whole-cell segmentation and tracking. This polishing work is crucial for generating ground truths for studies of natural variation.

All the evaluation and curation steps for the use cases are detailed in Materials and methods.

## Use-case 1: Assessing and improving segmentation quality of *Tribolium castaneum* embryos using unsupervised nuclei intensity metrics and iterative training on partial annotation

This use-case introduces two unsupervised metrics for assessing dataset quality without ground truth, demonstrating their application in evaluating segmentation performance. It also highlights the effectiveness of an iterative segmentation and training approach, where applying a Cellpose model trained on a curated subset significantly improves overall segmentation quality.

### Dataset description

This dataset features a 59-time-step light-sheet microscopy acquisition of labeled cell nuclei from a developing *Tribolium castaneum* embryo, containing over 7000 cells per time step. It includes two types of ground truth: a *gold truth* (GT), which is expert manual tracking of 181 nuclei over all time steps, and a *silver truth* (ST), generated through a complex pipeline combining the outputs of up to 16 top-performing algorithms selected from 39 submissions. The ST provides automated segmentation masks using a modified label fusion approach for improved accuracy in dense datasets. The GT and ST are independent and do not share common elements.

### Type of errors

This dataset serves as a benchmark in the fifth Cell Tracking Challenge (*Maška et al., 2023*) to evaluate and compare AI-based image segmentation methods and is, therefore, regarded as a reliable ground truth. The manual tracking GT is of high quality (expert annotation), but the automated ST is of lesser quality. To evaluate the segmentation quality of the dataset at first time point, the intensity image and the two ground truths (ST and GT) were simultaneously displayed in separate MorphoNet channels (*Figure 3b*). Systematic visual inspection identified 12/181 GT points without corresponding nucleus in the ST data; eight misplaced segmentations where the GT point is not inside the segmentation; 56 over segmentations and many suspicious nuclear shapes.

### Error identification

Using the *Match* plugin, each GT position was automatically associated with a corresponding nucleus in the ST, leaving 11 out of 181 positions unassigned (*Figure 3b*), indicating approximately 6% of missed nuclei in the ST. To identify inaccurately shaped segmentations, we used two signal intensity-based

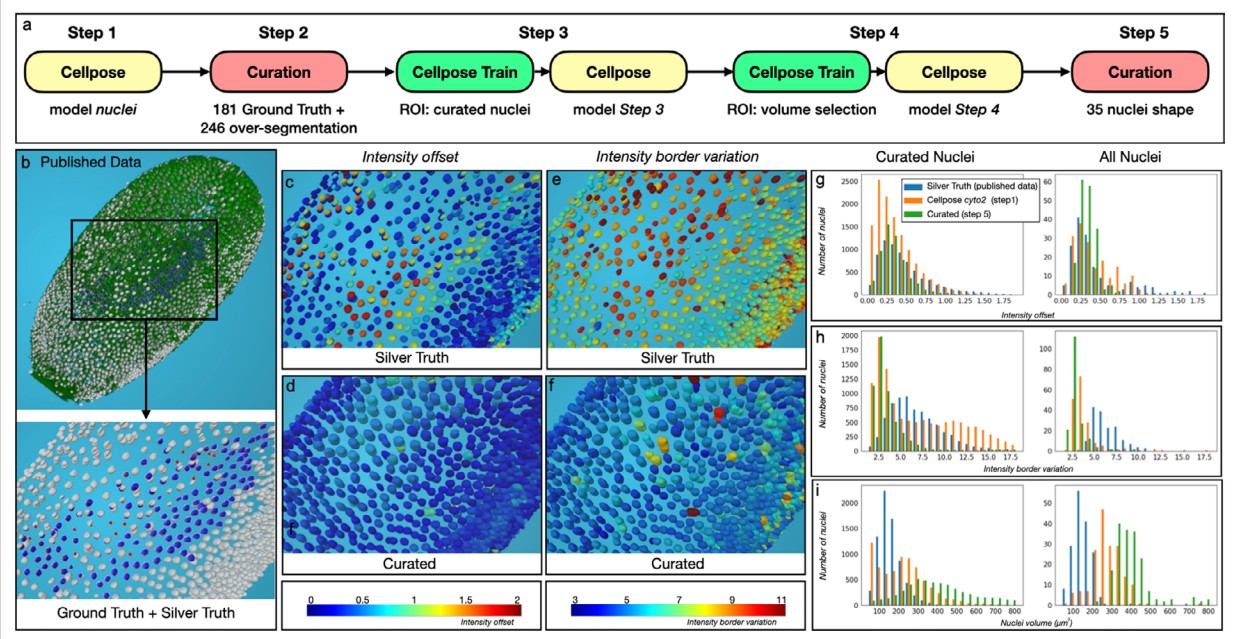

**Figure 3.** Unsupervised quality assessment and curation of the *Tribolium castaneum* embryo. (**a**) The five steps of the curation pipeline. (**b**) View of the original intensity images (in green) of the first time step of the published data (*Maška et al., 2023*). Both Ground Truth channel (GT, red point) and Silver Truth (ST, white nucleus segmentation) are shown at the top. Blue segmentation corresponds to the match between ST and GT. Bottom, zoom at the GT region (ROI) without the intensity images. (**c**) Projection for the ST of the distance between the gravity center of the intensity inside the segmentation and the centroid of the segmentation. Color bar at the bottom of **d**. (**d**) Same as **c** for the curated pipeline. (**e**) Projection for the ST of the deviation of the intensity at the border of the segmentation. Color bar at the bottom of **f**. (**f**) Same as **e** for the curated pipeline. (**g**) Comparative histogram of the intensity_offset property distribution between the ST, the Step 1 and the Step 5 for the 181 curated nuclei (left) and the whole image (right). (**h**) Same as **g** for the distribution of the intensity_border_variation property. (**i**) Same as **g** for the distribution of the nuclei volume property.

The online version of this article includes the following figure supplement(s) for figure 3:

**Figure supplement 1.** Illustration of properties *intensity_border_variation* and *intensity_offset for 2 nuclei*.

**Figure supplement 2.** Analysis of two automatically computed intensity properties.

**Figure supplement 3.** Distribution of three properties for the *Tribolium* reconstruction.

MorphoNet metrics. The first metric measures the distance between the geometric center of a segmented object and the center of mass of the signal intensity (*Figure 3—figure supplement 1* and Materials and methods for the full properties description), with 32 of the 56 over-segmented nuclei falling into the upper quartile of this metric's distribution (*Figure 3c and d*, *Figure 3—figure supplement 2*). The second metric evaluates deviations in the signal intensity along the segmentation boundary, with 29 out of 56 over-segmented nuclei identified in the upper quartile (*Figure 3c–f*, *Figure 3—figure supplement 1*). These errors were visualized in 3D by projecting both metrics onto each nucleus, enabling spatial identification of potential segmentation issues (*Figure 3—figure supplement 2*).

### Error correction

Due to the high number of errors in the published ST, a de novo nucleus segmentation pipeline was created in MorphoNet with minimal manual intervention. The five-step process (as detailed in Materials and methods UC1 Curated pipeline, *Video 1*) included initial segmentation using the standard Cellpose nuclei model, manual curation of 181 GT nuclei, iterative training of a custom

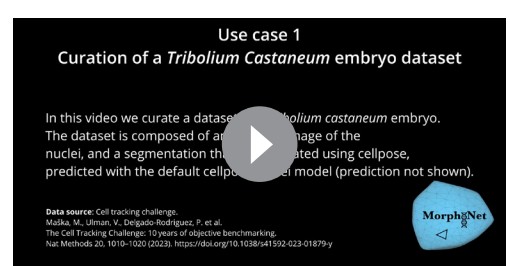

**Video 1.** *Tribolium castaneum*. The movie shows how to train a Cellpose model using a curated sub-part of a dataset, and how to fine-tune models with successive training on specific nuclei using image properties.
https://elifesciences.org/articles/106227/figures#video1

Cellpose model, and refinement of segmentation using geometric properties. Key steps involved correcting segmentation errors with various plugins, extending the Cellpose nuclei model using curated data, and refining non-convex shapes. The pipeline significantly improved segmentation quality compared to the published ST and standard Cellpose model (*Figure 3i*). Analysis of the curated nuclei revealed a more heterogeneous volume distribution, fewer over-segmentations, and better alignment of intensity-based metrics (*Figure 3g and h*, *Figure 3—figure supplement 3*), showing substantial improvement over the original segmentation pipeline. We also created a new set of manually segmented ground truth cells (see Materials and methods UC1 ground truth protocol), distributed across the ranges of both *intensity_offset* and *intensity_border_variation*. Using this reference, we evaluated the published and curated segmentations (*Figure 3—figure supplement 2h–j*), confirming that our pipeline both reduces segmentation errors and better matches expert annotations. Moreover, the analysis revealed a clear correlation between the unsupervised metrics and segmentation accuracy. These results validate the utility of our metrics and demonstrate that the iterative pipeline yields higher-quality segmentations than the published dataset.

## Use case 2: Evaluating membrane segmentation quality using *smoothness* metrics and simplifying a segmentation workflow for *Patiria miniata* starfish embryo

This use case thus introduces new unsupervised morphological metrics (the *smoothness* property) to assess segmentation quality and demonstrates that using a custom-trained 3D Cellpose model via graphical interfaces allows extending segmentation to areas of the dataset with lower intensity/noise ratio. Visualization of metrics, such as smoothness and cell volumes simplify the identification of cells that need further manual curation.

### Dataset description

This dataset comprises 300 3D stacks from time-lapse, two-channel confocal imaging of twelve *Patiria miniata* wild-type and compressed embryos, captured between the 128- and 512 cell stages (*Barone et al., 2024*). Fluorescent labeling highlights cell membranes and nuclei, though imaging was limited to about half of each embryo due to poor penetration and the use of low magnification air objectives. The published whole-cell segmentation (see Materials and methods UC2 Original workflow) relied on a complex, multi-step workflow adapted from the CartoCel (*Andrés-San Román et al., 2023*) requiring advanced bioanalysis expertise. For this use case, a particularly challenging dataset - a compressed wild-type embryo at the 512 cell stage - was selected for evaluation and curation.

### MorphoNet segmentation with the advanced cellpose plugin

We explored cellpose's potential to improve segmentation quality in challenging regions of a 3D dataset. Initial segmentation using the Cyto2 model identified 1187 cells, with a bimodal size distribution and 939 cells smaller than 1000 voxels (*Figure 4i*). The Deli plugin, which eliminates these small cells (*Figure 4j*), brought the number of cells down to 211 (*Figure 4j*); however, analysis with the *smoothness* property (*Figure 4g*) revealed rough cell surfaces. To address this, we further extend the training of the *Cyto2* model on the published 3D database (*Barone et al., 2024*) (see Materials and methods UC2 Training Protocol workflow, *Video 2*). The retrained model produced 284 cells with, again, a bimodal size distribution (*Figure 4i*) which was corrected with the *Deli* plugin. This significantly simpler pipeline produced smooth cell segmentations (*Figure 4h*), with a size distribution comparable to the published ground truth and a high Intersection over Union (IoU) score (*Figure 4—figure supplement 1*). While 2 cells were missing, and 11 under-segmented compared to the published data (*Figure 4f*), this new approach recovered 48 additional cells (*Figure 4e*) while simplifying the segmentation process compared to the original pipeline.

## Use case 3: Interactive targeted segmentation using cellpose for resolving under-segmentation in *Arabidopsis thaliana* shoot apical meristem whole-cell segmentation

This use case demonstrates the power of interactive selection with targeted segmentation methods that can resolve under-segmentation issues in complex datasets.

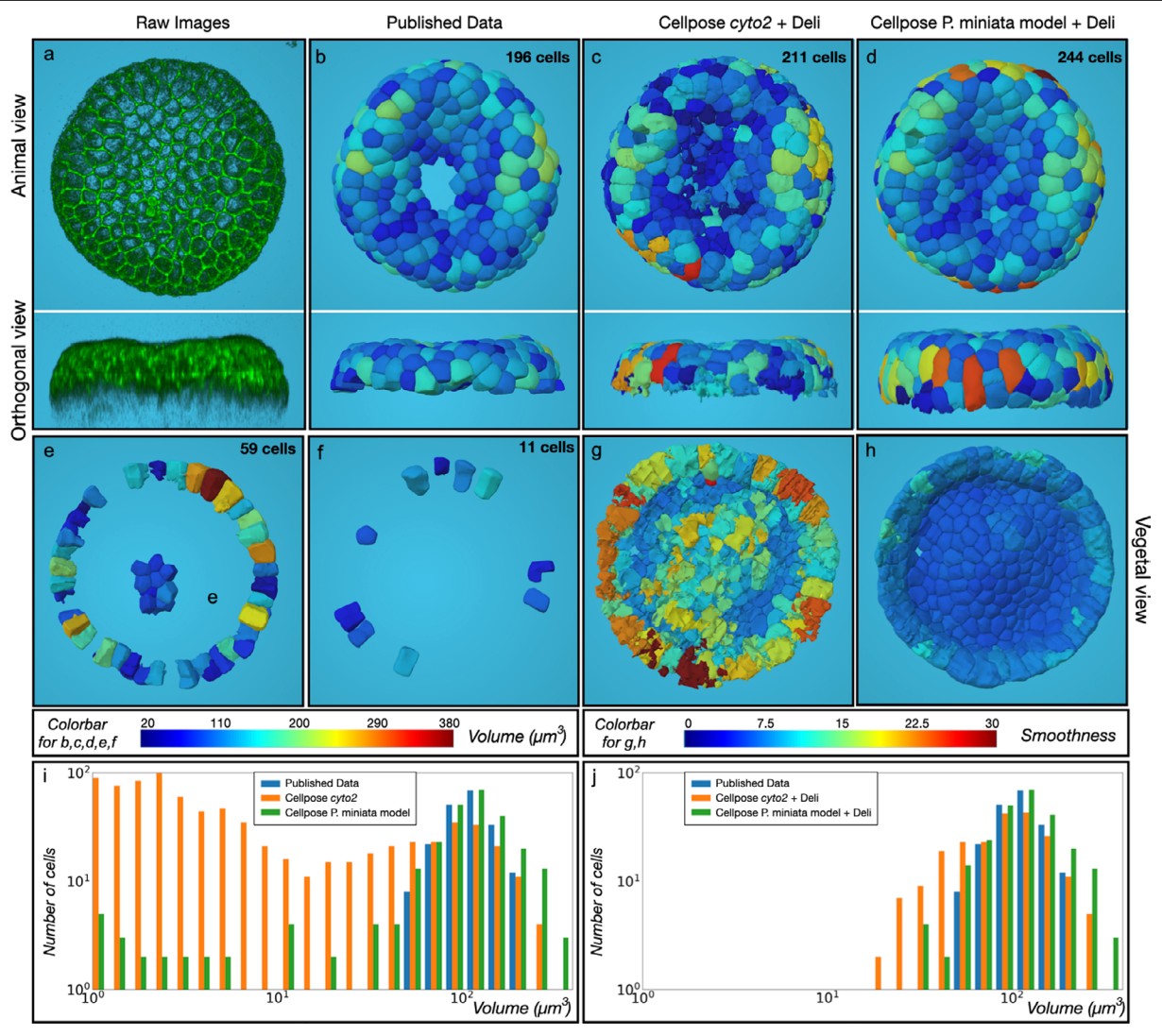

**Figure 4.** Starfish whole-cell segmentation using Cellpose. (**a**) Animal and Lateral view of the maximum intensity projection from the published dataset (*Barone et al., 2024*). (**b**) Segmented ground truth image. (**c**) Result of the segmentation using Cellpose Cyto2 model followed by the removal of small cells (<1000 voxels) with the Deli plugin (**d**) Result of a Cellpose segmentation with a model trained on *P. miniata* dataset followed by the Deli plugin. (**e**) New cells created (different between **b** and **d**) (**f**) Under segmentation and missing cells generated by **d** compared to **b**. (**g**) Vegetal view of **c** with smoothness representation. (**h**) Opposite view of **d** with smoothness representation. (**i**) Cell size distribution in the published segmentation (**b**), Cellpose cyto2 model (**c**), and Cellpose with *P. miniata* model (**d**). (**j**) Identical as **i** after application of the Deli plugin. Colors in **b-f** represent cell volume in µm³. Colors in **g**, **h** represent Smoothness.

The online version of this article includes the following figure supplement(s) for figure 4:

**Figure supplement 1.** Evaluation of Segmentation in *Patiria miniata* starfish embryo.

## Dataset description

This dataset (*Willis et al., 2016*) consists of a 19-time steps time-lapse confocal acquisition of a live *Arabidopsis thaliana* shoot apical meristem with fluorescently-labeled cell membranes. Cell numbers ranged from 600 cells at the first time point to around 1800 cells at the last (*Figure 5a*). We use the last time step (19) with the higher number of cells to illustrate the curation procedure.

## Error detection

The analysis of the published segmentation uncovered multiple under- and over-segmentation errors in the deep cell layers, caused by poor image quality in the inner regions of the meristem (*Figure 5b*). These errors hindered accurate cell tracking over time. Observing the size homogeneity of shoot apical

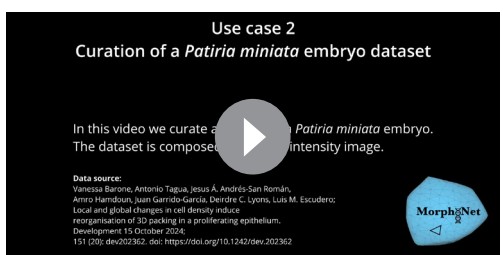

**Video 2.** *Patiria miniata*. The movie shows how to create a dataset, how to use an already existing custom Cellpose model and how to easily correct usual Cellpose segmentation errors.
https://elifesciences.org/articles/106227/figures#video2

meristem cells (*Figure 5d*), we hypothesized that larger-than-expected cells might indicate under-segmentation errors, while smaller cells could result from over-segmentation. By projecting the automatically computed cell volumes as a color map onto the segmentations, we identified a large under-segmented region containing 20 cells in the deep meristem layer (highlighted in red in *Figure 5c*), along with a few small over-segmented cells (*Figure 5i*).

## Error correction

The large number of fused cells in the under-segmented region prompted us to test whether our interactive Cellpose plugin could outperform the original MARS/ALT software (*Fernandez et al., 2010*). The standard pretrained *cyto2* model of Cellpose (*Pachitariu and Stringer, 2022*) produced numerous very small over-segmented cells (*Figure 5d*, *Figure 5—figure supplement 1*), suggesting it was not well-suited for this dataset (*Kar et al., 2022*). To improve performance, we used the high-quality MARS/ALT segmentations from the first ten time points and employed our Cellpose train plugin to extend the *cyto2* model training. We tested two training modalities: '2D' training, using only the XY planes, and '3D' training, incorporating XY, XZ, and YZ planes. The segmentations after additional 2D training still contained many over-segmented cells (*Figure 5f*), a problem that was significantly reduced with 3D training (*Figure 5g*). Despite this, the heterogeneity of signal intensities still caused over-segmentation in more superficial regions of the meristem. We thus targeted the Cellpose plugin only to the large under-segmented region (*Figure 5h*) and removed small over-segmented cells using the *Deli* segmentation correction plugin (*Figure 5i*). This targeted approach was fast and accurate (*Figure 5d*) and allowed us to preserve the accurate segmentations from the original published data while curating the identified errors. Notably, the curated result restored a unimodal and symmetric volume distribution (*Figure 5—figure supplement 1g*), consistent with the expected homogeneity of the shoot apical meristem (*Shapiro et al., 2015*), which was further confirmed by visual inspection (*Figure 5—figure supplement 1h–i*). With this approach, we successfully curated 20 under-segmented regions and generated 98 new cells (*Video 3*).

## Use-case 4: Improving segmentation quality using object editing plugins for *Caenorhabditis elegans* cell lineage

This use case illustrates how object editing plugins can be employed to improve segmentation quality in challenging datasets despite poor-quality nuclei labeling.

### Dataset description

This dataset (*Murray et al., 2008*) consists of a live 3D time-lapse confocal acquisition of a developing *Caenorhabditis elegans* embryo with fluorescently labeled cell nuclei (*Figure 6a*), over 195 time steps between the 4- and 362 cell stages. This dataset was included in the Cell Tracking Challenge (*Maška et al., 2023*) (see Materials and methods UC4 for the original workflow).

### Error detection

Although the manually curated published cell lineage appeared accurate (*Figure 6a*), some touching nuclei were elongated with flat vertical contact interfaces, likely due to poor imaging quality. To identify these segmentation artifacts systematically, we calculated the axis ratio for each segmented object and visualized these values directly on their surface meshes (*Figure 6b*). The distribution revealed two peaks: a sharp one around 2, containing most nuclei, and a broader one above 3. Nuclei with an axis ratio greater than 2.9 were flagged for potential correction.

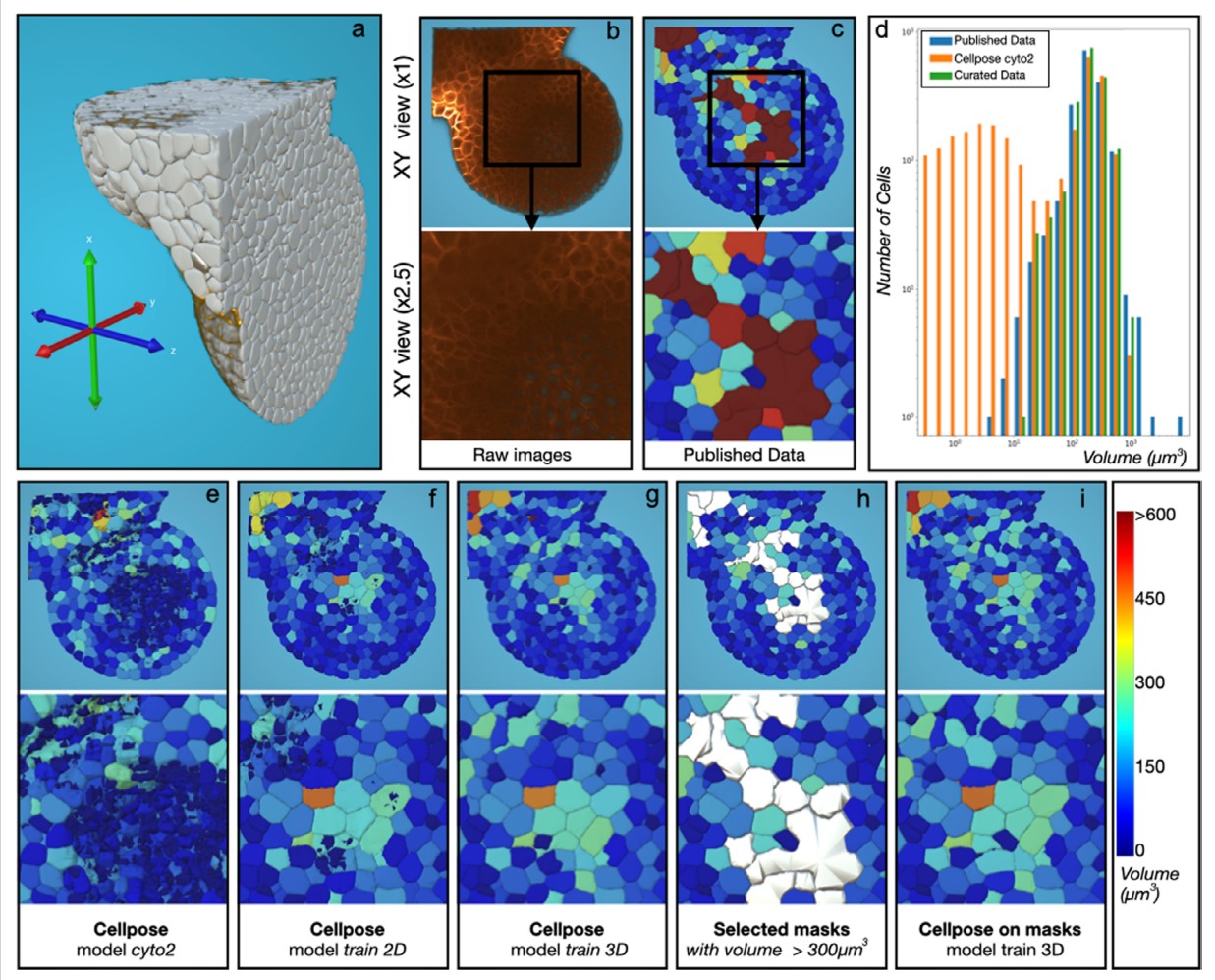

**Figure 5.** Curation of a segmented shoot apical meristem of *Arabidopsis thaliana*. Visualization of the time step n°19 for several types of curations using MorphoNet. (**a**) 3D view of an *Arabidopsis thaliana* shoot apical meristem (**Willis et al., 2016**). (**b**) Published 3D intensity images. (**c**) Published 3D segmentation of an *Arabidopsis thaliana* shoot apical meristem (**Willis et al., 2016**) obtained using MARS-ALT (**Fernandez et al., 2010**). (**d**) Comparative histogram based on the cell volume between the published segmentation (**c**), the result of the cyto2 prediction (**e**), and the final curated version (**i**) + Deli plugin for cells <1000 voxels. X and Y axes are in log scale. (**e**) Result of the Cellpose (**Stringer et al., 2021**) 3D MorphoNet plugin using the pretrained cyto2 model. (**f**) Result using the Cellpose 3D MorphoNet plugin using the model trained over the first 10 time steps with the Cellpose training plugin with the XY planes. (**g**) Result of the Cellpose 3D MorphoNet plugin using the model trained over the first 10 time steps with the Cellpose training plugin with each plane of the 3D images. (**h**) Selected masks larger than 300 µm³ in the published dataset (**c**). (**i**) Result of the Cellpose 3D MorphoNet on the selected masks (**h**) using the model trained over the ten first time steps with the Cellpose training plugin with each plane of the 3D images.

The online version of this article includes the following figure supplement(s) for figure 5:

**Figure supplement 1.** Comparison of the Cellpose models for the reconstruction shoot apical meristem of *Arabidopsis thaliana*.

## Error correction

We curated these errors for a specific time point by first applying a fusion correction plugin (*Figure 6f and g*) to all selected nuclei pairs with an axis ratio greater than 2.9 (*Figure 6e*). This was followed by a separation plugin (*Figure 6h*) that divides fused objects into distinct nuclei using a Gaussian mixture applied to intensity values. We initially addressed the more complex cases manually, which included one group of six fused nuclei, one group of four fused nuclei, and two groups of three. Next, we automatically curated the remaining nine simpler cases of two fused nuclei by selecting nuclei with a volume greater than 80 µm³. As expected, the axis ratio distribution of the curated set became unimodal and centered around 2 (*Figure 6b*), reflecting more regular nuclear shapes—an

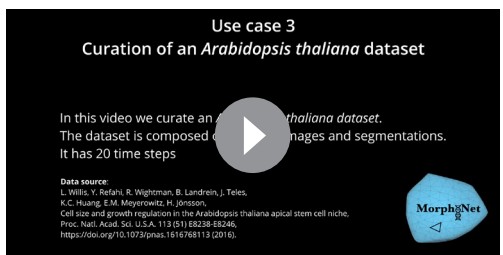

**Video 3.** *Arabidopsis thaliana*. The movie shows how to train a Cellpose model on several time steps of a 3D+t dataset, and how to use it to predict under-segmentations on a specific part of a dataset.
https://elifesciences.org/articles/106227/figures#video3

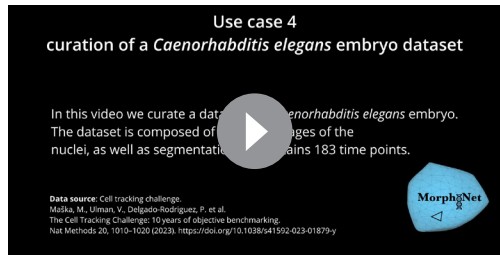

**Video 4.** *Caenorhabditis elegans*. The movie shows how to use the lineage and image properties to easily detect segmentation issues, and how to fix them in batches for a fast curation of dense datasets.
https://elifesciences.org/articles/106227/figures#video4

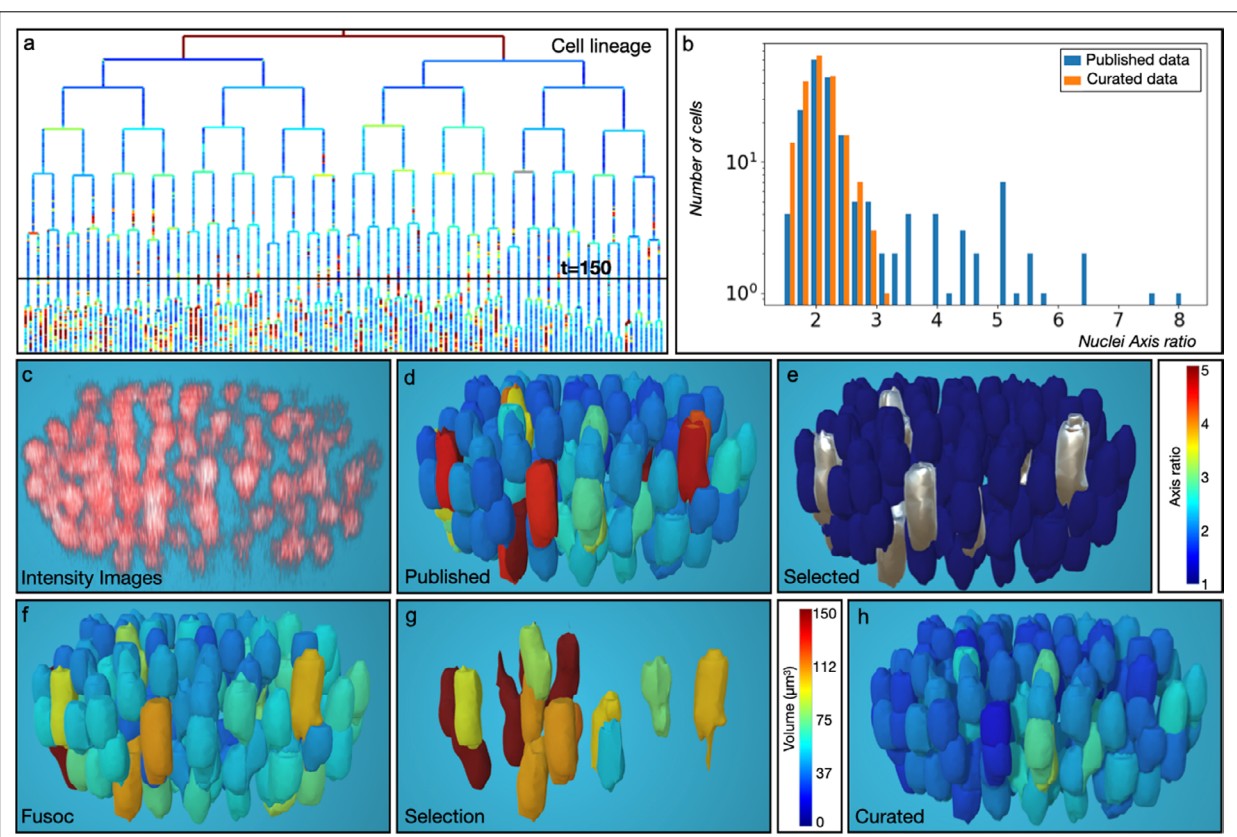

**Figure 6.** Curation of a *Caenorhabditis elegans* embryo dataset. (**a**) Cell lineage viewer of all time steps colored by clonal cells of the published segmented dataset (*Murray et al., 2008*). (**b**) Comparative histogram based on the axis ratio between the published data and the curation. (**c**) 3D View of the intensity images at t=150. (**d**) 3D View of the published segmented dataset at t=150. Colors represent the ratio of the longest axis on the shortest axis of the shape. (**e**) Automatic selection (in gray) of the nuclei with an axis ratio >2.9. (**f**) Result of the Fusoc plugin applied on all selected nuclei. Colors represent the nuclei volume in μm³. (**g**) Same as **f** but only previously selected nuclei are shown. (**h**) Result of the Gaumi plugin applied independently on regions fused with four, three or two nuclei. Colors represent the axis ratio as in **d**.

The online version of this article includes the following figure supplement(s) for figure 6:

**Figure supplement 1.** Manual Validation of Curation for the *Caenorhabditis elegans* Embryo.

improvement confirmed by visual inspection (*Figure 6—figure supplement 1*). The entire time step at t=150 was fully curated with just 5 plugin actions (*Figure 6h*, *Video 4*).

## Use-case 5: Enhancing cell lineage accuracy by detecting segmentation errors in *Phallusia mammillata* embryos

This use case showcases how MorphoNet can efficiently detect and correct segmentation errors in complex 3D+t datasets of developing ascidian embryos, enhancing the accuracy of cell lineage reconstructions and analysis of cell division timing variations.

### Dataset description

This dataset (*Guignard et al., 2020*) comprises 64 time steps of multi-view light-sheet microscopy of developing ascidian embryos (64–298 cell stages), segmented using the ASTEC algorithm (*Guignard et al., 2020*). Despite the high-quality segmentation and tracking, residual errors, such as multi-component cells, over- and under-segmentations, delayed, or missed divisions, and shape inaccuracies remained, with no tools available at the time of the publication to systematically validate or correct the 10,000 cell snapshots.

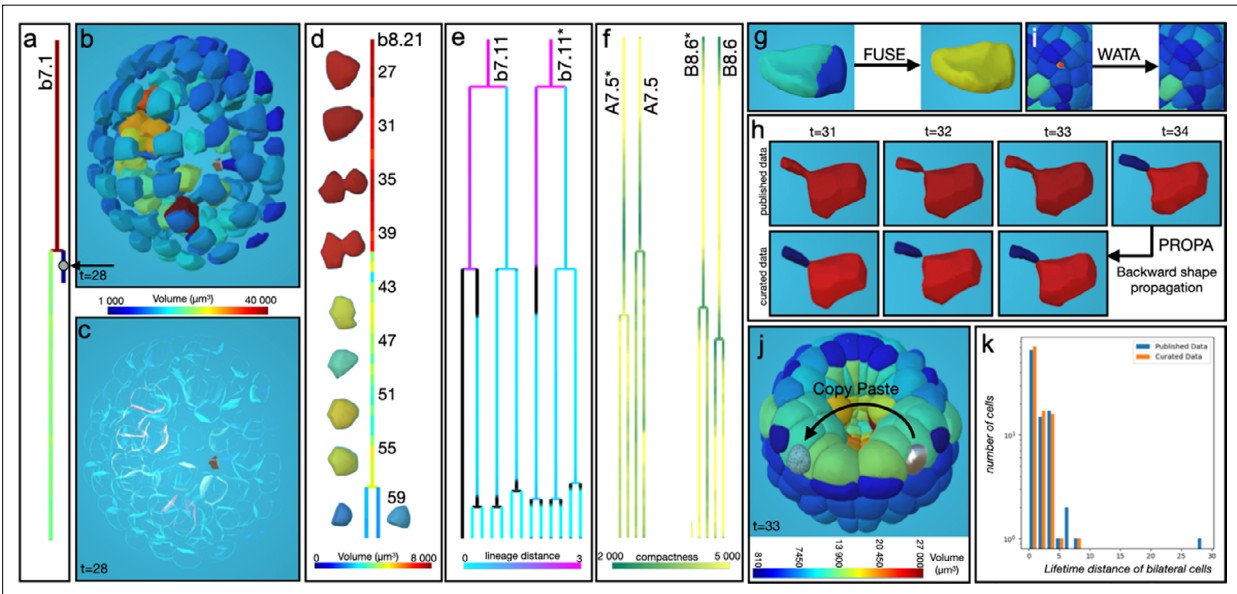

**Figure 7.** Curation of a *Phallusia mammillata* ascidian embryo named Astec-Pm9 in the publication (*Guignard et al., 2020*). (**a**) Cell lineage of the cell b7.1 after the execution of the Disco and Deli plugins on the published data. Projection of the volume property, the colormap between **b** and **c** represents the cell volume in µm³. (**b**) Scatter view of the corresponding segmented embryo described in **a**. at t=28 colored by cell volume. Colormap identical as **a**. (**c**) Same view as **b** with the activation of the 'highlight' mode which focuses on the selected cell and shows other cells in transparent colors. (**d**) Several snapshots of the cell b8.21 at different time points with its associated cell lineage. Colormap represents the cell volume in µm³ which points to a missing division. (**e**) Cell lineage of the bilateral cells b7.11 and b7.11*. Color bar shows the lineage distance between the bilateral symmetrical cells. Black region represents snapshots with no matches between bilateral symmetrical cells. (**f**) Cell lineage of the bilateral cells a7.5 and b8.6. Color bar shows the compactness property. The property highlights that the delay of division between A7.5* and A7.5 is due to an under-segmented error of A7.5*. B8.6 and B8.6* have expected behavior. (**g**) Result of the Fuse plugin applied on an over-segmented cell. (**h**) Top line: Several snapshots of B7.7* cell under-segmented (between time point 31 and 33) from the original segmented embryo. Bottom: result of the Propa plugin applied backward from time t=34 where both cells are well separated. (**i**) Example of the result of the Wata plugin from a manual added seed (red dot) in the empty space. (**j**) Result of the Copy-Paste plugin from the selected cell (in gray) on the left side of the embryo to the right side (the new cell appears with a mesh shader in blue). Colormap represents cell volume in µm³. (**k**) Comparison of the lifetime of bilateral symmetrical cells. The X-axis shows the number of in-time points separating the division of bilateral cell pairs. The Y-axis corresponds to the number of cells (in log).

The online version of this article includes the following figure supplement(s) for figure 7:

**Figure supplement 1.** Cell lineage following curation of a *Phallusia mammillata* embryo.

## Error detection

MorphoNet was used to polish the cell lineage to a sufficient level to study natural variation in cell division timing. The main challenge was identifying rare errors within a cell lineage that appeared accurate by visual inspection. To address common lineage residual errors, such as missing or delayed divisions and broken lineage links, we developed a set of segmentation error identification metrics. These metrics were visualized by projecting their results onto the lineage using an interactive viewer linked to the 3D dataset (*Figure 7a and b*). All identified errors stemmed from segmentation issues rather than tracking, and their correction was streamlined by the bidirectional connection between the lineage and embryo representation windows, allowing direct navigation between the two.

State-of-the-art segmentation methods often produce systematic errors, such as very small segmented objects, which we identified using geometric properties like cell volume (e.g. cell a7.1, *Figure 7a*). Property projection, combined with scatter visualization, facilitates the identification of cells even in dense, interior regions (*Figure 7b*), while the highlight mode enhances this process by isolating the cell of interest (*Figure 7c*). Occasional missing or false divisions disrupt accurate cell histories. During early ascidian embryogenesis, stable cell volumes allowed volume projections onto the lineage to identify rapid variations, revealing missed divisions (*Figure 7d*).

Ascidian embryos exhibit bilateral symmetry, with homologous cells dividing almost synchronously. A l*ineage distance* property (*Guignard et al., 2020*) revealed missed divisions (e.g. b7.11* in *Figure 7e*). To distinguish segmentation errors from natural variability, a *compactness* property was used to detect inaccuracies in division timing by tracking cell rounding during mitosis (e.g. *Figure 7f*).

Using MorphoNet, an expert identified approximately 20 isolated cell lineage errors in a dataset of 185 cell snapshots in just 2 hr.

## Error correction

The 41 multi-component labels were separated using the *Disco* plugin, identifying seven potential cells via the *volume* property analysis (*Figure 7b*). The others, considered as small artifacts, were merged with neighboring cells sharing the largest surface area using the *Deli* plugin. Division timing issues often stem from over-segmentation, under-segmentation, or missing cells. Over-segmentations were resolved with the *Fuse* plugin (*Figure 7g*), while under-segmentations were corrected using *Propagate segmentation* plugins by tracing back from the first accurate segmentation of sister cells to their mother's division point (*Figure 7h*). For missing cells, seeds were added with seed generator plugins and segmented using local seeded watershed algorithms (*Figure 7i*). If imaging quality was too poor, the bilaterally symmetrical cell served as a mirror-image proxy, using the Copy-Paste plugin to replicate, rotate, and scale the cell to the missing position (*Figure 7j*). A total of 185 errors were corrected in approximately eight hours using 264 MorphoNet plugin actions, resulting in a more accurate estimation of natural variability in cell division timing (*Figure 7k*, *Figure 7—figure supplement 1*). This duration includes the full workflow performed by a single user: loading the dataset, exploring the cell lineage, identifying segmentation or division errors, and applying corrective plugins. The reported actions include both exploratory attempts and validated corrections, combining user interaction with computation time (*Video 5*).

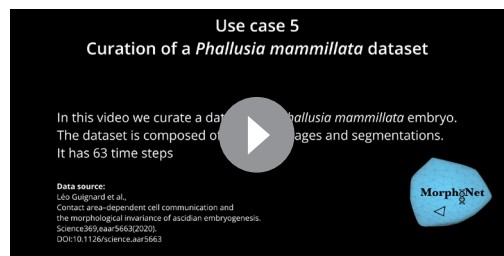

**Video 5.** *Phallusia mammillata*. This movie shows how to identify and fix several issues on a segmented dataset, using the lineage viewer and a large array of plugins. It shows how to fix large curation errors using a couple of actions only.

https://elifesciences.org/articles/106227/figures#video5

## Discussion

Recent advancements in optical time-lapse microscopy allow for the 3D capture of dynamic biological processes but face challenges in automating the segmentation and tracking of large, complex datasets. Residual segmentation errors in time-lapse datasets disrupt data interpretation and hinder long-term cell tracking. Moreover, accurate, high-quality 3D data is critical for training next-generation AI-based segmentation tools, yet the availability of such datasets remains limited.

To address these challenges, **MorphoNet 2.0** was developed as a standalone application for reconstructing, evaluating, and curating 3D and 3D+t datasets. Building on its previous web-based version, MorphoNet 2.0 integrates voxel images with meshed representations, leveraging both Unity Game Engine and Python for enhanced interactivity and processing power. Key features include tools for assessing segmentation quality through unsupervised metrics (e.g., volume, smoothness, and temporal stability), automated error detection, and visualization of segmented objects alongside raw intensity images.

Conventional image curation tools struggle with the complexities of interacting with 3D voxel images. In contrast, MorphoNet 2.0 introduces a novel approach using dual representation layers, enabling efficient biocuration by isolating problem areas and significantly accelerating dataset refinement. MorphoNet 2.0 is user-friendly, open-source, and supports both scientific discovery and AI training by producing high-precision 3D datasets and enabling reproducible, scalable data analysis.

By showcasing five use-cases of fluorescence datasets in which cell membranes or nuclei were labeled, we demonstrated that the tool's high versatility and user-friendliness enables biologists without programming skills to efficiently and intuitively detect and handle a broad range of errors (under-segmentation, over-segmentation, missing objects, lineage errors).

MorphoNet has a high potential to adapt to evolving datasets and segmentation challenges. First, its open-source Python plugin architecture fosters community-driven improvements. These could target cellular datasets as exemplified by the five use-cases presented. They could also open MorphoNet to other imaging modalities, including multi-modal datasets combining, for instance, fluorescence and electron microscopy. Additional plugins could accommodate new AI tools or automate the training of segmentation models through data augmentation, feature selection, or hyperparameter optimization. Plugins for widely used platforms, such as Napari or Fiji will also broaden the user-base and interoperability of the tool, as would also the development of flexible export options to integrate MorphoNet outputs with other analytical pipelines or visualization software. The MorphoNet platform could be further extended through the creation of a centralized repository for community-developed plugins, the organization of MorphoNet-based bio-image analysis challenges to stimulate community engagement, and the provision of curated datasets to serve as benchmarks for testing and validating new segmentation algorithms. To support these developments over time, we rely on institutional support from our host laboratories and research organizations, and we will seek additional funding through dedicated research grants. We also aim to foster open-source contributions, develop training materials, and provide user support to ensure long-term adoption and sustainability.

Curation could also be improved by the introduction of cloud-based, multi-user capabilities to enable experts to work on the same dataset simultaneously. For now, web browsers impose constraints on the use of computing resources, which could be lifted in the near future, for example, through the development of WebGPU.

Curation capabilities will likely also be enhanced by the implementation of adaptive machine learning models that integrate past user corrections to suggest or automate future edits. In the context of 3D segmentation tasks, manual expertise and curation for voxel-wise segmentation is labor-intensive and expensive. Full annotation of large datasets may thus not be feasible. Partial annotations allow datasets to be created with less time and resources while still providing valuable information. Including algorithms, such as Sketchpose (*Cazorla et al., 2025*) could leverage weakly-supervised learning to generalize from the partially annotated data and infer segmentation patterns in the unlabeled portions of the dataset, a strategy we initiated with the Tribolium dataset. This will make it possible to train models on diverse datasets without the burden of full annotations.

By addressing major challenges in 3D and 3D+t dataset assessment and curation, MorphoNet 2.0 provides a versatile platform for improving segmentation quality and generating reliable ground truths. Its user-friendliness, adaptability, and extensibility position it as a valuable tool for advancing quantitative bio-image analysis, with significant potential for enhancement and broader application in the future.

## Materials and methods

### Tools comparison

MorphoNet (MN) is a platform specifically designed for end users, requiring no programming skills. Its standalone version is a portable, code-free application, similar in spirit to Fiji (FJ).

MN provides interactive visualization of segmentations and supports both segmentation creation (via a range of plugins) and curation through a broad set of image processing tools. Unlike FJ and Napari (NP), which typically operate at the voxel level, MN performs curation exclusively at the object level. This object-centric approach enables faster and more intuitive editing of complex 3D segmentations.

MN also supports cell tracking and lineage visualization. It includes plugins for both automatic lineage generation and manual curation, similar to FJ's TrackMate plugin and NP's btrack (the latter requiring some coding knowledge).

MN offers built-in tools for visualizing and exporting morphological properties of segmented objects. In contrast to FJ, MN can directly map these properties onto segmentations using color-coded overlays, and can also project them onto lineage trees in its dedicated viewer.

Deep learning-based tools like Cellpose and StarDist are natively embedded in MN's standalone application, supporting both inference and model training with no additional installation. In comparison, FJ includes only a StarDist plugin, while NP supports both tools but typically requires command-line installation and environment setup.

Finally, all MN plugins are bundled and maintained by the development team, ensuring integration and stability. In contrast, plugin ecosystems in FJ and NP are open, allowing users to freely develop, install, and share new tools with the community.

### Curated and detailed pipeline for use cases

#### UC 1: *Tribolium castaneum* embryo cell nuclei segmentation

##### Curated pipeline

Due to the high number of errors in the published ST, we decided to generate de novo nucleus segmentation using MorphoNet. We defined a pipeline in five steps, which required only a few manual curations. Step 1- We started by launching the *Cellpose prediction* plugin using the pretrained standard *nuclei* model. Step 2- The nuclei of this segmentation corresponding to the published GT (181 nuclei) were curated with various plugins. Eight segmentations with wrong shape were recomputed using the *Binarize* plugin. One missing nucleus was added with the *BinCha* plugin which creates a new segmentation from the selected GT using a binary threshold on intensity image. Twelve segmentation errors (which were wrong segmentation boundaries between adjacent nuclei, and over-segmentations) could be corrected by using combinations of the *Fuse*, *Gaumi,* and *Delete* plugins. Also, using the intensity properties, we visually identified 248 over-segmented nuclei outside of the GT, that were easily corrected with the *Fuse* plugin. Step 3. We developed a *Cellpose Train* plugin with a user-friendly interface, which we used to extend pretrained Cellpose models (here *nuclei*) using XY, XZ, and YZ plane information. We added an option to train a model on selected objects within a region of interest of the image. We extended the *nuclei* model by an additional training step restricted to the 181 curated GT cells and used this new model to predict a new segmentation on the entire image. Step 4- Finally, based on this new segmentation, we automatically select nuclei with a volume between 1000 and 10,000 voxels and perform a new Cellpose train using this selection only. Step 5- To achieve a perfect shape for each GT-associated nuclei, we identified 35 nuclei with non-convex shapes using the *convexity* property, which we then refined using the *Shape Transform* plugin family (See Plugin list).

##### Manual segmentation protocol

To sample a representative set of segmentation quality levels, we used the histograms of the intensity_border_variation and intensity_offset properties computed from the original Cell Tracking Challenge dataset. For each metric, one cell was randomly selected from each of the 20 histogram bins, yielding a total of 40 cells. For each selected cell, a dilated bounding box was extracted from the original intensity image, and manual ground truth segmentation was performed using Napari's label editing tool. To detect over-segmentation, any segmented object other than the matched

one that overlapped the manual ground truth by at least 5% of its volume was classified as an over-segmentation.

## UC 2: Whole membrane segmentation of a *Patiria miniata* starfish embryo
### Original workflow

The published workflow started with the construction of a 3D Voronoi diagram based on cell centroids calculated from the nuclei channel acquisition, which was used to train a 3D ResU-Net (*Franco-Barranco et al., 2022*) on 14 consecutive stacks from a single embryo at the 256 cell stage. The predictions obtained with this trained model on wild-type and mutant embryos from the 128- to 512 cell stages were used as input for PlantSeg (*Wolny et al., 2020*), a watershed-based algorithm. Segmentation errors were corrected using custom Matlab scripts and the resulting segmentations were used to train a second 3D ResU-Net model, which was applied to the whole dataset. In compressed embryos, the authors choose to segment only 196 cells from a 512 cell stage embryo (*Figure 4b*), the signal intensity in the embryos' center and external circumferences being considered too low for precise and objective segmentation.

### Training Protocol for MorphoNet segmentation with the advanced Cellpose plugin

To train Cellpose 3D on the 300 *Patiria miniata* starfish embryos kindly provided by the authors of the paper (*Barone et al., 2024*), we split it into three subgroups: a training set corresponding to 80% of the database, a testing set (10%) and a validation set (10%). We then split 3D images into 2D slides in each direction (X, Y, Z). Cellpose has to preload data in memory to convert them in the native format before training. Due to inherent memory limitations, we had to split the training process into several trials. Each trial contains 5000 random images of the training set. We run 50 trials with each of them containing 50 epochs. We started the first trial using the *cyto2* model and then each successive trial was continued to train based on the model obtained in the previous one. We then evaluate each trial on the testing set and keep the one which give the best accurate segmentation.

## UC 4: *Caenorhabditis elegans* embryo cell nuclei segmentation
### Original workflow

The dataset contains 3D stacks of intensity images and two types of ground truth annotations. A silver truth (ST) automated segmentation of each nucleus using StarryNite (*Murray et al., 2006*). A gold truth (GT) manual annotation of the position of each nucleus over time, whose tracking was manually corrected with AceTree (*Boyle et al., 2006*).

## UC 5: *Phallusia mammillata* embryo whole-cell segmentation
### Error correction

The 41 labels (which should be cells) consisting of more than one connected component were disconnected using the *Disco* plugin. Visual inspection using the *volume* property of these objects identified seven potential cells. The others, considered as small artifacts, were removed by fusion to the cell neighbor they share the most surface with using the *Deli* plugin. This plugin requires the threshold value for the minimum cell volume, which can be manually found by specifically highlighting all small cells (*Figure 7b*).

Division timing issues typically arise from over-segmentation, under-segmentation or missed cells. Over-segmentations were easily corrected using the *Fuse* plugin (*Figure 7g*). Using the *Propagate segmentation* family plugins, under-segmentations were corrected by backwards propagation from the first accurate segmentation of the two sister cells in the dataset up to the precise moment of division of their mother (*Figure 7h*). Missing cells can be difficult to correct when signal intensity quality is poor. One correction option is to add a missing seed with the seed generator plugins. The expert can then apply one of several local seeded watershed algorithms to generate the missing cell segmentations (*Figure 7i*). When the imaging quality is too low for this strategy to succeed, we reasoned that the bilaterally symmetrical cell may be a good mirror-image approximation of the missing cell. We thus used the *Copy-Paste* plugin to copy a cell from her symmetric and to paste it, after appropriate rotation and scaling, where a cell is missing (*Figure 7j*).

Using a combination of using 264 MorphoNet plugin actions, 185 errors were corrected in 8 hr actions, leading to a better estimation of the natural variability in cell division timing (*Figure 7k*).

## Cellpose plugins

### Cellpose train plugin

The current version of Cellpose has limitations for 3D model training. It requires command-line or Python API usage, which is inaccessible for most experimental biologists. Additionally, Cellpose only trains on XY planes of 3D datasets, making it inefficient for anisotropic datasets with lower axial resolution, like the *Arabidopsis* dataset, resulting in over-segmented cells. These issues were addressed by developing a MorphoNet Cellpose Train plugin with a user-friendly interface that leverages information from the XY, XZ, and YZ planes, improving segmentation by fine-tuning pretrained models using 3D data. We added an option to make the image isotropic (by lineage interpolation) before launching the training which increases the model accuracy. Users have the possibility to train their own model on several time steps. Finally, in order to be able to use partial annotation, this plugin can be run only on selected masks, such as in the Tribolium dataset.

### Cellpose predict plugin

The Cellpose Predict plugin in MorphoNet is dedicated to the prediction of 3D segmented images. We implemented in MorphoNet an option to run each algorithm only on selected masks only (e.g., on cells containing segmented errors). Thus, this plugin can be run only on a subpart of the dataset (such as in *Arabidopsis* dataset). Cellpose generates multiple unconnected components with the same label which can be removed using the Disconnected Components option (Identical as the Disco Plugin). Applying the *cyto3* model on the *Arabidopsis* dataset generates 13293 objects with disconnected components (*Figure 5—figure supplement 1d and e*). We also add the possibility to automatically remove these small artifacts generated by Cellpose using an option to fuse or delete the objects below a given size (It will remove these objects if they are small artefacts in the background) (*Figure 5—figure supplement 1f*). This targeted approach is fastest, gives the best results and has the additional advantage of preserving the accurate cell segmentations.

## Properties

The majority of the region properties are computed from the scikit-image Python package as described here. Properties marked with a (v) are computed on the original image, whereas others are computed on an image re-scaled by its voxel size, to give appropriate physical measurements.

- volume (v): Corresponds to the *area* property of scikit-image. Area of the region, i.e., number of voxels in the region.
- volume-real: Volume property but scaled by the voxel-size to give a real physical volume **volume-bbox**: Corresponds to the *area_bbox* property of scikit-image. Area of the bounding box i.e., number of voxels of bounding box scaled by voxel-area.
- volume-filled: Corresponds to the *area_filled* property of scikit-image. Area of the region with all holes filled in.
- axis-major-length: The length of the major axis of the ellipse that has the same normalized second central moments as the region.
- axis-minor-length: The length of the minor axis of the ellipse that has the same normalized second central moments as the region.
- axis-ratio: Ratio of the longest axis over the smallest axis of the label (see *axis-major-length* and *axis-minor-length*).
- diameter (v): The mean diameter of the region. It is the mean of axis-major-length and axis-minor-length. IMPORTANT: this value is expressed in voxels, not physical size, so it can be of use for plugins that use voxel measurements, such as Cellpose, for instance. **Equivalent-diameter-area**: The diameter of a circle with the same area as the region.
- euler-number: Euler characteristic of the set of non-zero pixels. Computed as the number of connected components plus the number of holes, subtracted by the number of tunnels.
- extent: Ratio of pixels in the region to pixels in the total bounding box. Computed as *volume* divided by number of rows multiplicated by number of columns in bounding box **connected-neighbors**: number of connected other labels.

- convexity: distance to convexity. Computed as volume of the convex hull (smallest convex polygon that encloses the region) divided by the volume of the region.
- roughness: mean of the absolute values of the region minus the closing (dilation followed by erosion) of the region.
- compactness: computed as $\frac{region\ surface\ area^3}{36 \times \pi * volume^2}$
- smoothness: computed as $\frac{region\ surface\ area}{volume^{\frac{2}{3}}}$
- Intensity-max (v): Value with the greatest intensity in the region.
- Intensity mean (v): Value with the mean intensity in the region.
- Intensity-min (v): Value with the least intensity in the region.
- Intensity-border variation (v): the standard deviation of the intensity images only at the border of the segmentation.
- Intensity-offset (v): the Euclidean distance between the gravity center of the intensity images and the geometrical center of the segmentation.
- Lineage-distance: Requires a lineage property, as well as the *cell_name* property, containing the Conklin (*Child, 1906*) naming of cells. Compute the tree-edit distance between lineage trees (*Guignard et al., 2020*) of symmetrical cells.
- Bbox (v): bounding box of the region (tuple).

## Curation

The MorphoNet standalone application allows users to perform the 3D images curation using a new paradigm. The concept is based on a duality between the segmented image and the construction of 3D mesh objects for each individual label. While the meshes object are extremely powerful for 3D visualization and interaction, the 3D segmented image remains the most standard data type for image labels. The curation is performed in four steps:

1. Users identify their issues to curate using the viewer based on the interaction with the messages (using Lineage or Intensity Images)
2. Users identify and launch the most appropriate plugin to solve their issue
3. The plugin performs the curation directly inside the segmented image
4. MorphoNet automatically recomputes the modification of the meshes of the labeled which have been modified by the plugin and finally refresh the window

Each plugin performs a modification of the 3D segmented images. The modification is done locally on the backup of the segmented images stored inside the MorphoNet application, not on the original data. Finally, users can export their curation as new segmented images. Meshes can also be exported in standard formats, including OBJ, STL, and PLY.

MorphoNet includes a list of various 3D plugins described below. We also add the possibility for any bio-image analysis to simply develop, test, and add their own plugins (see *Supplementary file 1*). The Python environment already contains several libraries, such as NUMpy (*Harris et al., 2020*), scikit-image (*van der Walt et al., 2014*), scikit-learn (*Pedregosa, 2025*), TensorFlow (*Abadi, 2016*), and pytorch (*Paszke, 2019*).

## Curation plugins

MorphoNet already included a list of various 3D plugins which are classified by family types: De Novo Segmentation, De Novo Seed, Segmentation from Seeds, Segmentation Correction, Propagate Segmentation, Edit Temporal Links, and Shape Transform (*Figure 8*).

### De Novo segmentation

These plugins function without any previous segmentation and directly create a segmentation from the 3D intensity images. Alternatively, they can override any segmented image (or a part of) when used on already segmented images (Full online documentation).

### Stardist-predict: Perform nuclei segmentation on intensity images

This plugin uses intensity images of the nucleus from a local dataset to compute segmentations of the nucleus, using the 3D Stardist deep learning algorithm (*Weigert et al., 2020*). The default demo model of Stardist can be used, as well as custom models (which can be created by the Stardist-Train plugin).

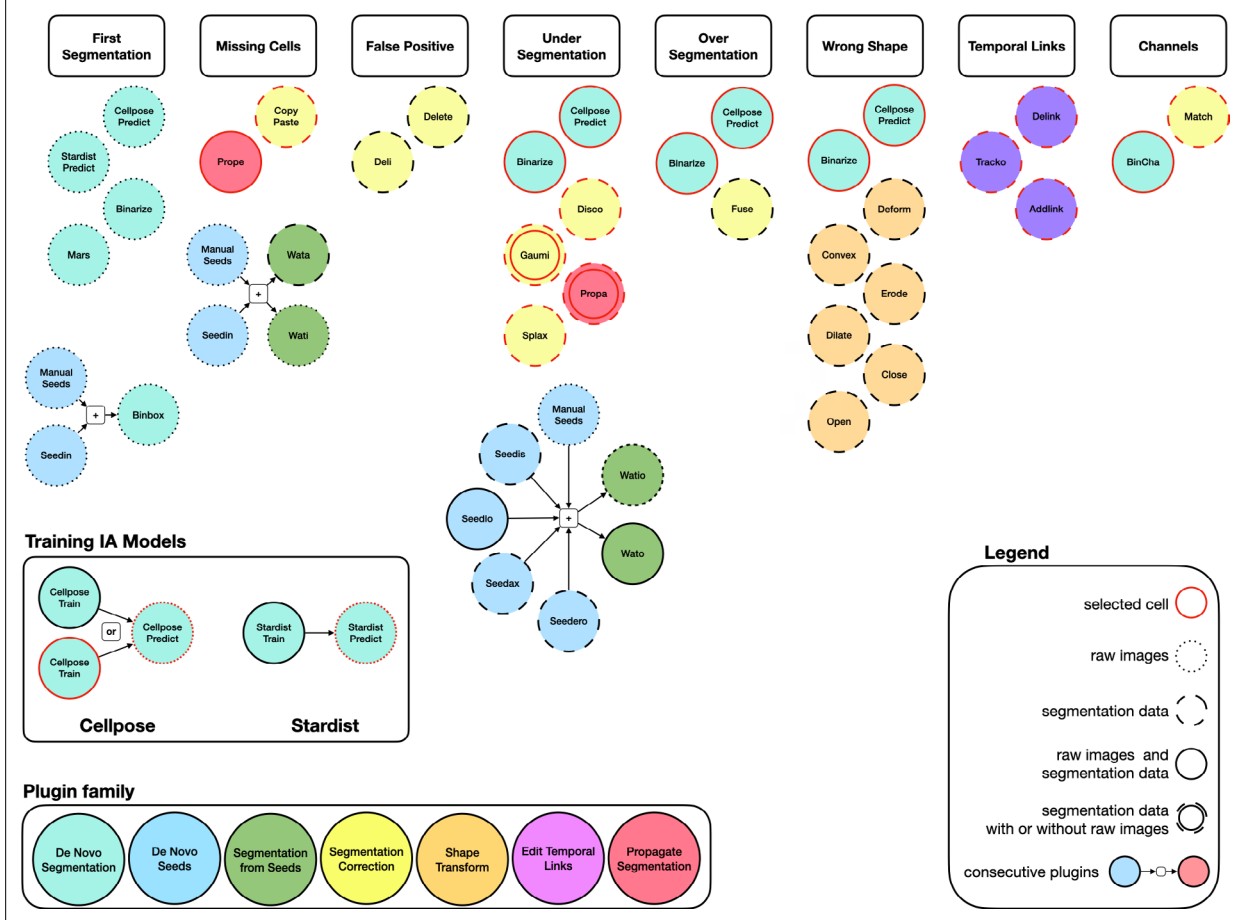

**Figure 8.** Representation of the MorphoNet Plugins organized by the corresponding functionalities according to segmentation issues. Color code represents the plugin family. Description of each plugin functionalities is accessible on the help web page: https://morphonet.org/help_curation#plugin_list.

### Stardist-train: Train the Stardist model for nuclei on your own data

This plugin allows users to train their own Stardist model of their 3D datasets. Using the 3D intensity image(s) with the corresponding segmentation(s), users can train their own models. The models can subsequently be used in the Stardist-Predict plugin to perform nuclei segmentation prediction on 3d intensity images (*Weigert et al., 2020*).

### Cellpose-predict: Perform membrane segmentation on intensity images

This plugin uses an intensity image of the membranes from a local dataset at a specific time point to compute a segmentation of the membranes, using the 3D Cellpose deep learning algorithm. Users can apply Cellpose on a 3D selected Mask to apply it to a part of a segmentation. By default, the plugin also disconnects all non-connex labels in the generated segmentation, and deletes all segmentations below a certain volume (in voxels), the same way it does in the *Disco* and *Deli* plugins. Each of these operations can be disabled with the plugin parameters.

### Cellpose-train: Perform membrane segmentation on intensity images

This plugin allows users to train their own model (from the models provided by Cellpose) on their own 3D datasets. With an intensity image and the corresponding segmentation, users can re-train a Cellpose model to obtain their own model, trained on their images. The model will be trained by using 2D images, which are the image stacks on the XY, XZ and YZ planes of the 3D images. Users can apply the plugin on a 3D selected Mask to train on a sub-part of a segmentation.

The model you output from this plugin can then be used in the Cellpose-Predict plugin, by inputting it in the pretrained_model parameter.

## Mars: Perform a seeded watershed segmentation on intensity images

This plugin uses an intensity image from a local dataset at a specific time point to perform a segmentation using a seeded watershed algorithm (*Pinidiyaarachchi and Wählby, 2006*).

## Binarize: Apply a threshold to intensity image

This plugin applies a threshold to the intensity image and then creates labels on each connected component above the threshold. This binarization can be applied on a given mask to run it on a subpart of the image.

## BinCha: Apply a binary threshold on the other channel from selected objects

This plugin creates a new segmentation from each mask of the selected objects using a binary threshold on intensity image in the desired channel. Alternatively, you can also do the thresholding on the centroid of each object, with a bounding box of input radius.

## BinBox: Binarize intensity images and label each object inside a bounding box

This plugin applies a threshold to intensity image on a bounding box and creates new labels on each connected component above the threshold.

## De Novo seed

These plugins automatically create seeds (e.g. 3D points within the 3D images), which can be then used by the plugins in the Segmentation from seeds family (Full online documentation).

## Seedio: Create seeds from minimum local intensity images on the selected objects

This plugin generates seeds that can be used in other plugins (mainly watershed segmentation). Users have to select objects (or label them) to generate seeds. Seeds are generated at the minima inside the selected object.

## Seedin: Create seeds from minimum local intensity images (without selected objects)

This plugin generates seeds that can be used in other plugins (mainly watershed segmentation). Seeds are generated at the minimum intensity where no segmentation labels are found (in the background).

## Seedis: Create seeds from the maximum distance to the border of the selected objects (without intensity images)

This plugin generates seeds that can be used in other plugins (mainly watershed segmentation). It computes the distance to the border of the selected objects and then extracts the maxima. N seeds are generated at the maximal distance inside the selected object (N being the number of seeds to generate), if the distance (between seeds) is above the threshold given by the min_distance parameter (in voxels, not physical size).

## Seedax: Create Seeds on the long axis of the selected objects (without intensity images)

This plugin generates seeds that can be used in other plugins (mainly watershed segmentation). The longest axis of the segmentation shape is computed, and then split in N segments (N being the number of seeds in parameter). Seeds are generated at the contact points of the segments. It requires the user to select or label objects on MorphoNet.

### Seedero: Create seeds from the erosion of selected objects (without intensity images)

This plugin generates seeds that can be used in other (mainly segmentation) algorithms. This plugin applies multiple erosion steps of each selected object, until objects can be separated into multiple unconnected parts. Then a seed is placed at the barycenter of each individual sub-part of the segmentation.

## Segmentation from seeds

These plugins can generate segmentations using seeds (mostly with a watershed algorithm *Najman and Schmitt, 1994*). Seeds can be either added by a plugin or manually on the interface (Full online documentation).

### Watio: Perform a watershed segmentation on intensity images on selected objects

This plugin creates new objects using a watershed algorithm from seed generated using a plugin or manually placed in the MorphoNet Viewer inside selected objects.

The watershed algorithm generates new objects based on the intensity image and replaces the selected objects. If the new generated objects are under the volume threshold defined by the user, the object is not created.

### Wati: Perform a watershed segmentation on intensity images (without selected objects)

This plugin creates new objects using a watershed algorithm from seed(s) generated or placed in the MorphoNet Viewer. The watershed algorithm generates new objects using the intensity image for each seed that is in the background. If the new generated objects are under a volume threshold defined by the user, the object is not created.

### Wato: Perform a watershed segmentation on selected objects (without intensity images)

This plugin creates new objects using a watershed algorithm from seed(s) generated using a plugin or placed in the MorphoNet Viewer inside selected objects. The watershed algorithm generates new objects using the segmentation image for each seed and replaces the selected objects. If the new generated objects are under the volume threshold defined by the user, the object is not created.

### Wata: Perform a watershed segmentation (without intensity images and without selected objects)

This plugin creates new objects using a watershed algorithm from seed(s) generated using a plugin or placed in the MorphoNet Viewer. The watershed algorithm generates new objects using the segmentation image for each seed inside a box that is not in another object. If the new generated objects are under the volume threshold defined by the user, the object is not created.

## Segmentation correction

A various list of plugins to perform actions at the level of the selected objects (fusion, deletion, split, copy-paste, etc.) (Full online documentation).

### Fuse: Fuse the selected objects

This plugin performs the fusion of selected objects into a single one. Multiple selected objects will be fused together at the current time point. If objects are labelled it will apply a fusion between all objects sharing the same label (for each individual time points).

### Gaumi: Split the selected objects using probability distribution

This plugin calculates the Gaussian mixture model probability distribution on selected objects in order to split them into several objects which will replace the selected ones.

### Disco: Split the selected objects for unconnected objects

This plugin can be used to split any object made up of several unconnected sub-objects.

### Splax: Split the selected objects in the middle of a given axis

This plugin splits any selected objects into two new objects in the middle of one of the given image axes.

### lete: Delete the selected objects

This plugin removes any selected objects from the segmented images. Users can select objects at the current time point, or label any objects to delete at several time points. The background value (usually 0) will replace the object voxels inside the segmented image.

### Deli: Delete any objects below a certain size

This plugin removes all objects that are under a certain volume (in voxel count) from the segmented images.

### Copy-paste: Copy a selected object and apply a transformation to the copy

This plugin gives the possibility to copy an object (a segmented cell, for example) and paste it at another time step and/or another location. The object can be moved, rotated, and scaled on the MorphoNet interface.

### Match: Matching objects across multiple channels

This plugin allows you to match the elements of the same object across several channels. It will give the selected objects a matching label in the segmented image. Can be used in batch by labeling objects together with label groups.

## Shape transform

These plugins allow the user to change the shape of objects, with Morphological operators (Erode, Dilate, etc.) and even manually (Full online documentation).

### Dilate: Dilate the selected objects

This plugin performs the dilation morphological operator on each individual selected object.

### Erode: Erode the selected objects

This plugin performs the erosion morphological operator on each individual selected object.

### Open: Open the selected objects

This plugin performs the opening morphological operator on each individual selected object.

### Close: Close the selected objects

This plugin performs the closing morphological operator on each individual selected object.

### Convex: Make selected objects convex

This plugin computes the convex voxel hull of each individual selected object.

### Deform: Apply a manual deformation on the selected object

This plugin allows the user to manually deform a selected object using mesh deformation. This plugin must be used with the mesh morphing menu and its various tools. The mesh morphing menu allows the user to manually deform a selected object of your dataset by applying various transformations (move vertices, extrude, hollow,...) with the mouse pointer. Once the user is satisfied with the deformation, this plugin computes the transformation(s) applied to the mesh to the segmented image in the dataset, and regenerates the mesh object using this segmented data.

## Propagate segmentation

These plugins allow the user to propagate a good segmentation at a given time to other time points (Full online documentation).

### Propa: Propagate selected eroded objects through time on the next object (with or without intensity images)

This plugin propagates labeled objects at the current time point through time. It requires applying a specific label to the objects to propagate at the current time (named the source objects) And then labeling the corresponding objects on which to propagate at the next or previous time points (named the destination objects). The plugin can be executed forward (source objects are at the beginning of the time range) or backward (source objects are at the end of the time range). The source objects are eroded until they fit into destination objects at the next time point, and then a watershed is computed. The watershed algorithm can be computed using an intensity image by checking the 'use intensity' box. The new objects created by a watershed will replace the destination objects.

This plugin propagates labeled objects at the current time point through time, filling empty space. The selected object(s) (named the source objects) will be propagated forward in time or backward in time by choosing the appropriate time_direction parameter. The source objects are then copied to the segmentation of the next/previous time (the intersection with the rest of the segmentation is removed), and finally a watershed algorithm is applied, using the corresponding intensity image. New objects are created in the background of the segmentation.

## Edit temporal links

These plugins can create or modify temporal information in order, for example, to curate a cell lineage tree (Full online documentation).

### Addlink: Create temporal links between labeled objects

This plugin creates temporal links at several time points on objects sharing the same label. After the execution, the lineage property is updated.

### Delink: Delete temporal links on labeled objects

This plugin deletes temporal links at several time points on objects sharing the same label. After the execution, the lineage property is updated.

### Tracko: Create temporal links using the overlap between all objects

This plugin creates a complete object lineage using the maximum of overlap between objects. The overlap is calculated between the bounding box enveloping each object. After the execution, the lineage property is updated.

## Create a new plugin

It is possible for any user to develop their own curation plugins in Python, and integrate them in their version of the MorphoNet standalone application. Users who wish to develop their own plugins can do so by following a tutorial on the MorphoNet website help page: https://morphonet.org/help_api?menu=morphonetplot#plugins.

## Code availability

The MorphoNet platform is composed of several Open-Source repositories:

- The frontend 3D Viewer code and Unity3D project is available here: (https://gitlab.inria.fr/MorphoNet/morphonet_unity, *Laurent, 2025a*).
- The python API code, which contains the python backend of the standalone application is available here: (https://gitlab.inria.fr/MorphoNet/morphonet_api, *Laurent, 2025b*).
- The lineage viewer code and Unity3D project is available here: (https://gitlab.inria.fr/MorphoNet/morphonet_lineage, *Laurent, 2025c*).

## Acknowledgements

This work was supported by grants of the Occitanie Region (ESR-PREMAT-213) and by the French National Infrastructure France BioImaging (ANR-10-INBS-04) to EF, by the Cell-Whisper (ANR-19-CE13-0020), the scEmbryo-Mech (ANR-21-CE13-0046) ANR projects and the Fondation pour la Recherche Médicale (EQU202303016262) coordinated by PL. EF and PL were CNRS staff scientists. NF was an assistant professor at UM. KB was a UM Phd Student supported by the EpiGenMed Labex (ProjetIA-10-LABX-0012) and a post-doc funded by the scEmbryo-Mech project. BG,TL were contract CNRS engineers. AC was a UM Master student. We thank Christophe Godin, Vanessa Barone, Thibault de Villèle, and Volker Baecker for their valuable feedback and advice.

## Additional information

### Funding

| Funder | Grant reference number | Author |
|---|---|---|
| Occitanie Region | ESR-PREMAT-213 | Emmanuel Faure |
| Agence Nationale de la Recherche | ANR-10-INBS-04 | Emmanuel Faure |
| Agence Nationale de la Recherche | ANR-19-CE13-0020 | Patrick Lemaire |
| Agence Nationale de la Recherche | ANR-21-CE13-0046 | Patrick Lemaire |
| EpiGenMed Labex | ProjetIA-10-LABX-0012 | Patrick Lemaire |
| Fondation pour la Recherche Médicale | EQU202303016262 | Patrick Lemaire |

The funders had no role in study design, data collection and interpretation, or the decision to submit the work for publication.

### Author contributions

Benjamin Gallean, Tao Laurent, Conceptualization, Data curation, Software, Visualization; Kilian Biasuz, Conceptualization, Data curation, Validation, Visualization; Ange Clement, Noura Faraj, Software; Patrick Lemaire, Conceptualization, Supervision, Investigation, Methodology, Writing – review and editing; Emmanuel Faure, Conceptualization, Resources, Data curation, Software, Formal analysis, Supervision, Funding acquisition, Validation, Investigation, Visualization, Methodology, Writing – original draft, Project administration, Writing – review and editing

### Author ORCIDs

Kilian Biasuz (iD) https://orcid.org/0000-0002-0240-2754
Patrick Lemaire (iD) https://orcid.org/0000-0003-4925-2009
Emmanuel Faure (iD) https://orcid.org/0000-0003-2787-0885

Reviewer #2 (Public review): https://doi.org/10.7554/eLife.106227.3.sa1
Author response https://doi.org/10.7554/eLife.106227.3.sa2

## Additional files

### Supplementary files

MDAR checklist

Supplementary file 1. Summary of the benchmarking datasets and device performance evaluation for the MorphoNet standalone application.

Supplementary file 2. Overview of the MorphoNet documentation hub, organizing help resources by user profile and intended use cases.

## Data availability

The MorphoNet_Data repository (https://doi.org/10.6084/m9.figshare.30529745.v1) provides all datasets and resources associated with the MorphoNet 2.0 publication. It serves as an open archive supporting the reproducibility and reuse of the study's results. The repository includes: DATASETS - Original and curated imaging datasets used in MorphoNet 2.0. BENCHMARKING_DATASETS - Reference datasets employed for performance evaluation, as detailed in the benchmark documentation and supplementary materials. MODELS - Pre-trained Cellpose models specifically used for segmentation curation in each dataset. MOVIES - Demonstration videos illustrating the use cases of each dataset.

The following dataset was generated:

| Author(s) | Year | Dataset title | Dataset URL | Database and Identifier |
|---|---|---|---|---|
| Gallean B, Laurent T, Biasuz K, Clement A, Faraj N, Lemaire P, Faure E | 2025 | MorphoNet 2.0: An innovative approach for qualitative assessment and segmentation curation of large-scale 3D time-lapse imaging datasets | https://doi.org/10.6084/m9.figshare.30529745.v1 | figshare, 10.6084/m9.figshare.30529745.v1 |

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
