## [Editor Report · eLife Assessment]

This **important** work presents technical and conceptual advances with the release of MorphoNet 2.0, a versatile and accessible platform for 3D+T segmentation and analysis. The authors provide **compelling** evidence across diverse datasets, and the clarity of the manuscript together with the software's usability broadens its impact. Although the strength of some improvements is hard to fully gauge given sample complexity, the tool is a significant step forward that will likely impact many biological imaging fields.

---

## [Referee Report · Reviewer #2 (Public review)]

Summary:

This article presents Morphonet 2.0, a software designed to visualise and curate segmentations of 3D and 3D+t data. The authors demonstrate its capabilities on five published datasets, showcasing how even small segmentation errors can be automatically detected, easily assessed and corrected by the user. This allows for more reliable ground truths which will in turn be very much valuable for analysis and training deep learning models. Morphonet 2.0 offers intuitive 3D inspection and functionalities accessible to a non-coding audience, thereby broadening its impact.

Strengths:

The work proposed in this article is expected to be of great interest for the community, by enabling easy visualisation and correction of complex 3D(+t) datasets. Moreover, the article is clear and well written making MorphoNet more likely to be used. The goals are clearly defined, addressing an undeniable need in the bioimage analysis community. The authors use a diverse range of datasets, successfully demonstrating the versatility of the software.

We would also like to highlight the great effort that was made to clearly explain which type of computer configurations are necessary to run the different dataset and how to find the appropriate documentation according to your needs. The authors clearly carefully thought about these two important problems and came up with very satisfactory solutions.

Weaknesses:

Sometimes, it can be a bit difficult to assess the strength of the improvements made by the proposed methods, but this is not something the authors could easily address, given the great complexity of the samples

---

## [Author Response]

The following is the authors’ response to the original reviews.

**Reviewer #1 (Public review):**
The authors present a substantial improvement to their existing tool, MorphoNet, intended to facilitate assessment of 3D+t cell segmentation and tracking results, and curation of high-quality analysis for scientific discovery and data sharing. These tools are provided through a user-friendly GUI, making them accessible to biologists who are not experienced coders. Further, the authors have re-developed this tool to be a locally installed piece of software instead of a web interface, making the analysis and rendering of large 3D+t datasets more computationally efficient. The authors evidence the value of this tool with a series of use cases, in which they apply different features of the software to existing datasets and show the improvement to the segmentation and tracking achieved.While the computational tools packaged in this software are familiar to readers (e.g., cellpose), the novel contribution of this work is the focus on error correction. The MorphoNet 2.0 software helps users identify where their candidate segmentation and/or tracking may be incorrect. The authors then provide existing tools in a single user-friendly package, lowering the threshold of skill required for users to get maximal value from these existing tools. To help users apply these tools effectively, the authors introduce a number of unsupervised quality metrics that can be applied to a segmentation candidate to identify masks and regions where the segmentation results are noticeably different from the majority of the image.This work is valuable to researchers who are working with cell microscopy data that requires high-quality segmentation and tracking, particularly if their data are 3D time-lapse and thus challenging to segment and assess. The MorphoNet 2.0 tool that the authors present is intended to make the iterative process of segmentation, quality assessment, and re-processing easier and more streamlined, combining commonly used tools into a single user interface.

We sincerely thank the reviewer for their thorough and encouraging evaluation of our work. We are grateful that they highlighted both the technical improvements of MorphoNet 2.0 and its potential impact for the broader community working with complex 3D+t microscopy datasets. We particularly appreciate the recognition of our efforts to make advanced segmentation and tracking tools accessible to non-expert users through a user-friendly and locally installable interface, and for pointing out the importance of error detection and correction in the iterative analysis workflow. The reviewer’s appreciation of the value of integrating unsupervised quality metrics to support this process is especially meaningful to us, as this was a central motivation behind the development of MorphoNet 2.0. We hope the tool will indeed facilitate more rigorous and reproducible analyses, and we are encouraged by the reviewer’s positive assessment of its utility for the community.

One of the key contributions of the work is the unsupervised metrics that MorphoNet 2.0 offers for segmentation quality assessment. These metrics are used in the use cases to identify low-quality instances of segmentation in the provided datasets, so that they can be improved with plugins directly in MorphoNet 2.0. However, not enough consideration is given to demonstrating that optimizing these metrics leads to an improvement in segmentation quality. For example, in Use Case 1, the authors report their metrics of interest (Intensity offset, Intensity border variation, and Nuclei volume) for the uncurated silver truth, the partially curated and fully curated datasets, but this does not evidence an improvement in the results. Additional plotting of the distribution of these metrics on the Gold Truth data could help confirm that the distribution of these metrics now better matches the expected distribution.Similarly, in Use Case 2, visual inspection leads us to believe that the segmentation generated by the Cellpose + Deli pipeline (shown in Figure 4d) is an improvement, but a direct comparison of agreement between segmented masks and masks in the published data (where the segmentations overlap) would further evidence this.

We agree that demonstrating the correlation between metric optimization and real segmentation improvement is essential. We have added new analysis comparing the distributions of the unsupervised metrics with the gold truth data before and after curation. Additionally, we provided overlap scores where ground truth annotations are available, confirming the improvement. We also explicitly discussed the limitation of relying solely on unsupervised metrics without complementary validation.

We would appreciate the authors addressing the risk of decreasing the quality of the segmentations by applying circular logic with their tool; MorphoNet 2.0 uses unsupervised metrics to identify masks that do not fit the typical distribution. A model such as StarDist can be trained on the "good" masks to generate more masks that match the most common type. This leads to a more homogeneous segmentation quality, without consideration for whether these metrics actually optimize the segmentation

We thank the reviewer for this important and insightful comment. It raises a crucial point regarding the risk of circular logic in our segmentation pipeline. Indeed, relying on unsupervised metrics to select “good” masks and using them to train a model like StarDist could lead to reinforcing a particular distribution of shapes or sizes, potentially filtering out biologically relevant variability. This homogenization may improve consistency with the chosen metrics, but not necessarily with the true underlying structures.

We fully agree that this is a key limitation to be aware of. We have revised the manuscript to explicitly discuss this risk, emphasizing that while our approach may help improve segmentation quality according to specific criteria, it should be complemented with biological validation and, when possible, expert input to ensure that important but rare phenotypes are not excluded.

In Use case 5, the authors include details that the errors were corrected by "264 MorphoNet plugin actions ... in 8 hours actions [sic]". The work would benefit from explaining whether this is 8 hours of human work, trying plugins and iteratively improving, or 8 hours of compute time to apply the selected plugins.

We clarified that the “8 hours” refer to human interaction time, including exploration, testing, and iterative correction using plugins.

**Reviewer #2 (Public review):**
Summary:This article presents Morphonet 2.0, a software designed to visualise and curate segmentations of 3D and 3D+t data. The authors demonstrate their capabilities on five published datasets, showcasing how even small segmentation errors can be automatically detected, easily assessed, and corrected by the user. This allows for more reliable ground truths, which will in turn be very much valuable for analysis and training deep learning models. Morphonet 2.0 offers intuitive 3D inspection and functionalities accessible to a non-coding audience, thereby broadening its impact.Strengths:The work proposed in this article is expected to be of great interest to the community by enabling easy visualisation and correction of complex 3D(+t) datasets. Moreover, the article is clear and well written, making MorphoNet more likely to be used. The goals are clearly defined, addressing an undeniable need in the bioimage analysis community. The authors use a diverse range of datasets, successfully demonstrating the versatility of the software.We would also like to highlight the great effort that was made to clearly explain which type of computer configurations are necessary to run the different datasets and how to find the appropriate documentation according to your needs. The authors clearly carefully thought about these two important problems and came up with very satisfactory solutions.

We would like to sincerely thank the reviewer for their positive and thoughtful feedback. We are especially grateful that they acknowledged the clarity of the manuscript and the potential value of MorphoNet 2.0 for the community, particularly in facilitating the visualization and correction of complex 3D(+t) datasets. We also appreciate the reviewer’s recognition of our efforts to provide detailed guidance on hardware requirements and access to documentation—two aspects we consider crucial to ensuring the tool is both usable and widely adopted. Their comments are very encouraging and reinforce our commitment to making MorphoNet 2.0 as accessible and practical as possible for a broad range of users in the bioimage analysis community.

Weaknesses:There is still one concern: the quantification of the improvement of the segmentations in the use cases and, therefore, the quantification of the potential impact of the software. While it appears hard to quantify the quality of the correction, the proposed work would be significantly improved if such metrics could be provided.The authors show some distributions of metrics before and after segmentations to highlight the changes. This is a great start, but there seem to be two shortcomings: first, the comparison and interpretation of the different distributions does not appear to be trivial. It is therefore difficult to judge the quality of the improvement from these. Maybe an explanation in the text of how to interpret the differences between the distributions could help. A second shortcoming is that the before/after metrics displayed are the metrics used to guide the correction, so, by design, the scores will improve, but does that accurately represent the improvement of the segmentation? It seems to be the case, but it would be nice to maybe have a better assessment of the improvement of the quality.

We thank the reviewer for this constructive and important comment. We fully agreed that assessing the true quality improvement of segmentation after correction is a central and challenging issue. While we initially focused on changes in the unsupervised quality metrics to illustrate the effect of the correction, we acknowledged that interpreting these distributions was not always straightforward, and that relying solely on the metrics used to guide the correction introduced an inherent bias in the evaluation.

To address the first point, we revised the manuscript to provide clearer guidance on how to interpret the changes in metric distributions before and after correction, with additional examples to make this interpretation more intuitive.

Regarding the second point, we agreed that using independent, external validation was necessary to confirm that the segmentation had genuinely improved. To this end, we included additional assessments using complementary evaluation strategies on selected datasets where ground truth was accessible, to compare pre- and post-correction segmentations with an independent reference. These results reinforced the idea that the corrections guided by unsupervised metrics generally led to more accurate segmentations, but we also emphasized their limitations and the need for biological validation in real-world cases.

**Reviewer #3 (Public review):**
Summary:A very thorough technical report of a new standalone, open-source software for microscopy image processing and analysis (MorphoNet 2.0), with a particular emphasis on automated segmentation and its curation to obtain accurate results even with very complex 3D stacks, including timelapse experiments.Strengths:The authors did a good job of explaining the advantages of MorphoNet 2.0, as compared to its previous web-based version and to other software with similar capabilities. What I particularly found more useful to actually envisage these claimed advantages is the five examples used to illustrate the power of the software (based on a combination of Python scripting and the 3D game engine Unity). These examples, from published research, are very varied in both types of information and image quality, and all have their complexities, making them inherently difficult to segment. I strongly recommend the readers to carefully watch the accompanying videos, which show (although not thoroughly) how the software is actually used in these examples.

We sincerely thanked the reviewer for their thoughtful and encouraging feedback. We were particularly pleased that the reviewer appreciated the comparative analysis of MorphoNet 2.0 with both its earlier version and existing tools, as well as the relevance of the five diverse and complex use cases we had selected. Demonstrating the software’s versatility and robustness across a variety of challenging datasets was a key goal of this work, and we were glad that this aspect came through clearly. We also appreciated the reviewer’s recommendation to watch the accompanying videos, which we had designed to provide a practical sense of how the tool was used in real-world scenarios. Their positive assessment was highly motivating and reinforced the value of combining scripting flexibility with an interactive 3D interface.

Weaknesses:Being a technical article, the only possible comments are on how methods are presented, which is generally adequate, as mentioned above. In this regard, and in spite of the presented examples (chosen by the authors, who clearly gave them a deep thought before showing them), the only way in which the presented software will prove valuable is through its use by as many researchers as possible. This is not a weakness per se, of course, but just what is usual in this sort of report. Hence, I encourage readers to download the software and give it time to test it on their own data (which I will also do myself).

We fully agreed that the true value of MorphoNet 2.0 would be demonstrated through its practical use by a wide range of researchers working with complex 3D and 3D+t datasets. In this regard, we improved the user documentation and provided a set of example datasets to help new users quickly familiarize themselves with the platform. We were also committed to maintaining and updating MorphoNet 2.0 based on user feedback to further support its usability and impact.

In conclusion, I believe that this report is fundamental because it will be the major way of initially promoting the use of MorphoNet 2.0 by the objective public. The software itself holds the promise of being very impactful for the microscopists' community.
**Reviewer #1 (Recommendations for the authors):**
(1) In Use Case 1, when referring to Figure 3a, they describe features of 3b?

We corrected the mismatch between Figure 3a and 3b descriptions.

(2) In Figure 3g-I, columns for Curated Nuclei and All Nuclei appear to be incorrectly labelled, and should be the other way around.

We corrected the label swapped between “Curated Nuclei” and “All Nuclei.”

(3) Some mention of how this will be supported in the future would be of interest.

We added a note on long-term support plans

(4) Could Morphonet be rolled into something like napari and integrated into its environment with access to its plugins and tools?

We thank the reviewer for this pertinent suggestion. We fully recognize the growing importance of interoperability within the bioimage analysis community, and we have been working on establishing a bridge between MorphoNet and napari to enable data exchange and complementary use of the two tools. As a platform, all new developments are first evaluated by our beta testers before being officially released to the user community and subsequently documented. The interoperability component is still under active development and will be announced shortly in a beta-testing phase. For this reason, we were not able to include it in the present manuscript, but we plan to document it in a future release.

(5) Can meshes be extracted/saved in another format?

We agreed that the ability to extract and save meshes in standard formats was highly useful for interoperability with other tools. We implemented this feature in the new version of MorphoNet, allowing users to export meshes in commonly used formats such as OBJ or STL. Response: We thank the reviewer for this pertinent suggestion. We fully recognize the growing importance of interoperability within the bioimage analysis community, and we have been working on establishing a bridge between MorphoNet and napari to enable data exchange and complementary use of the two tools. As a platform, all new developments are first evaluated by our beta testers before being officially released to the user community and subsequently documented. The interoperability component is still under active development and will be announced shortly in a beta-testing phase. For this reason, we were not able to include it in the present manuscript, but we plan to document it in a future release.

**Reviewer #2 (Recommendations for the authors):**
As a comment, since the authors mentioned the recent progress in 3D segmentation of various biological components, including organelles, it could be interesting to have examples of Morphonet applied to investigate subcellular structures. These present different challenges in visualization and quantification due to their smaller scale.

We thank the reviewer for this insightful suggestion. We fully agree that applying MorphoNet 2.0 to the analysis of sub-cellular structures is a promising direction, particularly given the specific challenges these datasets present in terms of resolution, visualization, and quantification. While our current use cases focus on cellular and tissue-level segmentation, we are actively interested in extending the applicability of the tool to finer scales. We are currently exploring plugins for spot detection and curation in single-molecule FISH data. However, this requires more time to properly validate relevant use cases, and we plan to include this functionality in the next release.

Another comment is that the authors briefly mention two other state-of-the-art softwares (namely FIJI and napari) but do not really position MorphoNet against them. The text would likely benefit from such a comparison so the users can better decide which one to use or not.

We agreed that providing a clearer comparison between MorphoNet 2.0 and other widely used tools such as FIJI and Napari would greatly benefit readers and potential users. In response, we included a new paragraph in the supplementary materials of the revised manuscript, highlighting the main features, strengths, and limitations of each tool in the context of 3D+t segmentation, visualization, and correction workflows. This addition helped users better understand the positioning of MorphoNet 2.0 and make informed choices based on their specific needs.

Minor comments:L 439: The Deli plugin is mentioned but not introduced in the main text; it could be helpful to have an idea of what it is without having to dive into the supplementary material.

We included a brief description in the main text and thoroughly revise the help pages to improve clarity

Figure 4: It is not clear how the potential holes created by the removal of objects are handled. Are the empty areas filled by neighboring cells, for example, are they left empty?

We clarified in the figure legend of Figure 4.

Please remove from the supplementary the use cases that are already in the main text.

We cleaned up redundant use case descriptions.

Typos:L 22: the end of the sentence is missing.L 51: There are two "."L 370: replace 'et' with 'and'.L 407-408, Figure 3: panels g-i, the columns 'curated nuclei' and 'all nuclei' seem to be inverted.L 549: "four 4".
**Reviewer #3 (Recommendations for the authors):**
Dear Authors, what follows are "minor comments" (the only sort of comment I have for this nice report):Minor issues:(1) Not being a user of MorphoNet, I found that reading the manuscript was a bit hard due to the several names of plugins or tools that are mentioned, many times without a clear explanation of what they do. One way of improving this could be to add a table, a sort of glossary, with those names, a brief explanation of what they are, and a link to their "help" page on the web.

We understood that the manuscript might be difficult to follow for readers unfamiliar with MorphoNet, especially due to the numerous plugin and tool names referenced. To address this, we carried out a complete overhaul of the help pages to make them clearer, more structured, and easier to navigate.

(2) Figure 4d, orthogonal view: It is claimed that this segmentation is correct according to the original intensity image, but it is not clear why some cells in the border actually appear a lot bigger than other cells in the embryo. It does look like an incomplete segmentation due to the poor image quality at the border. Whether this is the case or if the authors consider the contrary, it should be somehow explained/discussed in the figure legend or the main text.

We revised the figure legend and main text to acknowledge the challenge of segmenting peripheral regions with low signal-to-noise ratios and discussed how this affects segmentation.

Small writing issues I could spot:Line 247: there is a double point after "Sup. Mat..".Line 329: probably a diagrammation error of the pdf I use to review, there is a loose sentence apparently related to a figure: "Vegetal view ofwith smoothness".Line 393 (and many other places): avoid using numbers when it is not a parameter you are talking about, and the number is smaller than 10. In this case, it should be: "The five steps...".Line 459: Is "opposite" referring to "Vegetal", like in g? In addition, it starts with lower lowercase.Lines 540-541: Check if redaction is correct in "...projected the values onto the meshed dual of the object..." (it sounds obscure to me).Lines 548-549: Same thing for "...included two groups of four 4 nuclei and one group of 3 fused nuclei.".Line 637: Should it be "Same view as b"?Line 646: "The property highlights..."?Line 651: In the text, I have seen a "propagation plugin" named as "Prope", "Propa", and now "Propi". Are they all different? Is it a mistake? Please, see my first "Minor issue", which might help readers navigate through this sort of confusing nomenclature.Line 702: I personally find the use of the term "eco-system" inappropriate in this context. We scientists know what an ecosystem is, and the fact that it has now become a fashionable word for politicians does not make it correct in any context.

We thank the reviewer for their careful reading of the manuscript and for pointing out these writing and typographic issues. We corrected all the mentioned points in the revised version, including punctuation, sentence clarity, consistent naming of tools (e.g., the propagation plugin), and appropriate use of terms such as “ecosystem.” We also appreciated the suggestion to avoid numerals for numbers under ten when not referring to parameters, and we ensured consistency throughout the text. These corrections improved the clarity and readability of the manuscript, and we were grateful for the reviewer’s attention to detail.